# BARRON APPROXIMATION AND LOCALLY OPTIMAL WEIGHT DENSITIES FOR SHALLOW NEURAL NETWORKS

## ABSTRACT

Mean field theories provide optimization results for shallow neural networks by analyzing the weight distribution in infinite width limits. Corresponding results for finite sized networks are obtained by particle approximations, for which sharp quantitative bounds are still an open problem. In this paper, we consider a modified mean field loss, which allows a more fine grained control over finite sized networks. We prove convexity and equidistribution properties, which directly lead to Barron type approximation results for networks sampled from local loss minimizers. We demonstrate that particle approximations of the new loss function naturally lead to gradient descent methods with dropout type regularization.

## 1 INTRODUCTION

Continuum limits have proven to be helpful for both the optimization and approximation theory of shallow neural networks. While there are several possible variants, in this paper we consider the limit

$$\frac{1}{m}\sum_{i=1}^{m} a_i \sigma(w_i) \quad \rightarrow \quad \int a_\pi(w)\sigma(w)\,d\pi(w), \tag{1}$$

for probability measure $\pi$ and some function $a_\pi(w)$. The function $\sigma\colon \mathcal{W} \to \mathcal{H}$ maps weights from domain $\mathcal{W} \subset \mathbb{R}^{d_\mathcal{W}}$ to a Hilbert space $\mathcal{H}$, with the main motivation coming from shallow neural networks given by $\mathcal{H} = L_2$ and $\sigma(w)(x) = ReLU(v^T x - b)$ for $w = (v, b)$.

The limit (1) is sometimes referred to as a two-timescale limit (Marion and Berthier, 2023; Takakura and Suzuki, 2024), if the outer $a_i$ or $a_\pi(w)$ are chosen optimally. This differs slightly from the more common limit $\int a\sigma(w)\,d\pi(a, w)$ in mean filed theory, where $a$ is not a function but included in the measure's domain.

### 1.1 APPROXIMATION THEORY

We want to approximate a target function $f$ as close as possible by a neural network. For its construction, we assume that $f$ is given as a continuum limit

$$f = \int a_\pi(w)\sigma(w)\,d\pi(w), \tag{2}$$

and then we find the finite network by sampling $w_i \sim \pi$, effectively reversing the arrow in (1). This leads to error bounds of the form

$$\inf_{\boldsymbol{a},\boldsymbol{w}} \left\|\frac{1}{m}\sum_{i=1}^{m} a_i \sigma(w_i) - f\right\| \le \mathbb{E}_{\boldsymbol{w}\sim\pi^m}\left\|\frac{1}{m}\sum_{i=1}^{m} a_\pi(w_i)\sigma(w_i) - f\right\| \lesssim m^{-1/2}\,\|f\|_{B(\pi)}, \tag{3}$$

for some norm $\|\cdot\|_{B(\pi)}$ defined in Section 2.2. This norm matches Barron or variation norms for a careful choice of $\pi$. This choice, however, is not available in practice and the error bound (3) works for many other distributions. Indeed, for any probability distribution $\pi'$ that has a density $d\pi/d\pi'$ with respect to $\pi$ we can set $a_{\pi'} := a_\pi d\pi/d\pi'$ and obtain $f = \int a_{\pi'}(w)\sigma(w)\,d\pi'(w)$, as well as the corresponding approximation errors. Despite the similarity in error bounds, the choice of $\pi$ has

profound practical implications. For unfavourable choices, the norm $\|f\|_{B(\pi)}$ may be excessively large or even infinite.

This problem can already be seen in the simplest examples in one spacial dimension where shallow ReLU networks correspond to splines and the choice of $\pi$ to a (random) adaptive placement of spline knots. Similar problems also arise in the selection of local refinement in adaptive finite element methods (DeVore, 1998; Cohen and Mirebeau, 2009). In those classical theories, the best choices are achieved by equidistributing local errors or local smoothness. We argue that this principle still applies to shallow neural networks, with equidistribution defined by

$$|a_\pi(w)| \, \|\sigma(w)\| = \text{constant in } w. \tag{4}$$

As we see in Section C.2, the left hand side can be understood as a local smoothness indicator, relative to the distribution $\pi$. The condition also ensures that all samples $w_i \sim \pi$, receive the same outer weight $a_\pi(w_i)$, up to a scaling factor. With equidistribution , the norm $\|f\|_{B(\pi)}$ reduces to the Barron or variation norm found in current approximation results. The main objective of this paper is to show that such theoretical choices are attained by practical optimization methods.

## 1.2 Mean Field Limit

Mean field theory leverages the continuum limit (1) to understand the optimization of neural networks, starting from the objective

$$\min_\pi \left\| \int a_\pi(w)\sigma(w) \, d\pi(w) - f \right\|^2, \tag{5}$$

or more commonly with the alternative continuum limit $\int a\sigma(w) \, d\pi(a, w)$. This problem is convex in $\pi$ and optimized by Wasserstein gradient flow or Langevin dynamics. Implications for finite networks are achieved by replacing $\pi$ with a particle approximation $\pi \approx \frac{1}{m} \sum_{i=1}^m \delta_{w_i}$ for which Wasserstein gradient flow matches the gradient flow of the finite networks.

While the methods converge to global optima, it is not clear that all global optima are good ones: They achieve zero loss and thus match the exact representation (2). But as we have seen, there are many such representations with severely different performance of finite approximations. The very recent paper Takakura and Suzuki (2024), which we became aware of after completion of this paper, controls this choice by an extra penalty term, similar to the $\|f\|_{B(\pi)}$ norm above. We choose an alternative route without penalties but a different loss function.

## 1.3 New Contributions

Instead of the standard mean field loss (5), we optimize the expected sampling error in the approximation bound (3) directly

$$\min_\pi \mathbb{E}_{\boldsymbol{w} \sim \pi^m} \left\| \frac{1}{m} \sum_{i=1}^m a_\pi(w_i)\sigma(w_i) - f \right\|^2. \tag{6}$$

Unlike mean field theory, the expectation is outside of the norm, which ensures that the loss is non-zero and provides a direct control over finite sized networks. Due to the outer product measure $\pi^m = \pi \otimes \cdots \otimes \pi$ and the $\pi$-dependent function $a_\pi$, this optimization problem may appear overly difficult. Nonetheless, the main results of this paper show that it has several favourable properties:

1. Theorem 3.1: The optimization problem (6) is convex over admissible probability distributions $\pi$.

2. Lemmas 3.2, 3.4: Local minimizers $\pi$ satisfy the *equidistribution* property (4)

$$|a_\pi(w)| \, \|\sigma(w)\| = \text{constant + perturbations}, \qquad \text{for all } w, \pi - \text{a.s.}$$

3. Theorem 3.3, 3.5: Local minimizers $\pi$ achieve the approximation errors

$$\mathbb{E}_{\boldsymbol{w} \sim \pi^{\mathfrak{m}}} \left\| \frac{1}{\mathfrak{m}} \sum_{i=1}^{\mathfrak{m}} a_\pi(w_i)\sigma(w_i) - f \right\|^2 \leq \frac{4}{\mathfrak{m}} \left[ |f|_{B(\delta,\epsilon)}^2 + \pi(\delta) \right] + \epsilon^2.$$

for all $\mathfrak{m} \in \mathbb{N}$, where $|\cdot|_{B(\delta,\epsilon)}$ is a stable variant of classical Barron norms and $\epsilon, \delta$ perturbation terms dependent on $a_\pi(w)$.

4. Section 4: As for the standard mean field loss, Wasserstein gradient flow matches gradient flow of finite networks. For corresponding gradient descent methods, the outer expectation naturally leads to a dropout regularization.

We consider two variants for these results, depending on the definition of $a_\pi(w)$. The first is an idealized scenario, with a streamlined analysis, where $a_\pi$ is given as the Radon-Nikodym derivative of an ideal distribution. The second is a more practical choice, where $a_\pi$ is defined as the optimal coefficients of finite networks, averaged over off-diagonal weights.

## 1.4 LITERATURE REVIEW

**Approximation** The approximation theory of neural networks seeks to understand how well a target function $f$ can be approximated by a neural network. Universal approximation theorems (Cybenko, 1989; Hornik et al., 1989; Barron, 1993; Zhou, 2020; Lu et al., 2017; Hanin and Sellke, 2017) show that this is possible to arbitrary accuracy.

Since these early results, lots of effort has been spent on a more fine grained picture that seeks quantitative error bounds given various different smoothness properties of the target function and architectures of the networks. For Sobolev or Besov smooth targets, classical approximation methods like finite elements or splines are effective and neural networks match their performance (Gribonval et al., 2022; Gühring et al., 2020; Opschoor et al., 2020; Li et al., 2019; Suzuki, 2019). If one allows discontinuous weight assignments, neural networks can achieve higher approximation rates that classical methods (Yarotsky, 2017; 2018; Yarotsky and Zhevnerchuk, 2020; Daubechies et al., 2022; Shen et al., 2019; Lu et al., 2021). Sobolev and Besov smoothness is not suitable for high dimensional targets and with more tailored smoothness assumptions in Barron and variation norms, neural networks achieve dimension independent approximation rates (Bach, 2017; Klusowski and Barron, 2018; Weinan et al., 2022; Li et al., 2020; Siegel and Xu, 2020; 2022a; Bresler and Nagaraj, 2020; Parhi and Nowak, 2021; Unser, 2023; Siegel and Xu, 2024). More information is contained in the reviews (Pinkus, 1999; DeVore et al., 2021; Weinan et al., 2020; Berner et al., 2022).

Approximation theorems state the existence of good neural network approximation, but usually do not include a corresponding training mechanism. Early results for gradient descent trained networks (Jentzen and Riekert, 2022; Ibragimov et al., 2022; Drews and Kohler, 2022; Kohler and Krzyzak, 2022; Gentile and Welper, 2024; Welper, 2024a; 2025) rely on the convex outer layer, or lazy linearized training regimes and therefore cannot match the full approximation power known in theory. Barron smoothness approximation results can be achieved by greedy algorithms (Siegel and Xu, 2022b; Siegel et al., 2023) which reduce training to another non-convex sub-problem.

**Landscape Analysis** Properties of the training objective are studied in landscape analysis. With strong assumptions or over-parametrization, local minima are global minima (Soudry and Carmon, 2016; Kawaguchi, 2016; Nguyen and Hein, 2017; Ge et al., 2018; Du and Lee, 2018; Soltanolkotabi et al., 2019; Venturi et al., 2019; Kawaguchi et al., 2019; Kawaguchi and Huang, 2019). In different regimes this is not true (Swirszcz et al., 2017; Safran and Shamir, 2018; He et al., 2020; Ding et al., 2022; Jentzen and Riekert, 2024), or local minima are path connected (He et al., 2020). Approximation properties of critical points, are studied in Welper (2024b). The reference achieves sharp optimization bounds for one dimensional problems by optimizing the network weights. In comparison, we optimize the weight distribution and obtain results in high dimensions.

**Lazy Training Regime** For very wide networks, the gradient descent training dynamics are dominated by linearization at the initial value. This leads to gradient descent convergence based on the neural tangent kernel, originally introduced in (Jacot et al., 2018; Li and Liang, 2018; Allen-Zhu et al., 2019; Du et al., 2019b;a), and then further developed in e.g. (Zou et al., 2020; Arora et al., 2019a;b; Su and Yang, 2019; Lee et al., 2019; Song and Yang, 2019; Zou and Gu, 2019; Kawaguchi and Huang, 2019; Chizat et al., 2019; Oymak and Soltanolkotabi, 2020; Ji and Telgarsky, 2020; Nguyen and Mondelli, 2020; Bai and Lee, 2020; Cao and Gu, 2020; Chen et al., 2021; Song et al., 2021; Lee et al., 2022; Gentile and Welper, 2024; Welper, 2024a; 2025; Welper and Keene, 2025). Due to the inherent linearization, these approaches do not match the network's full potential, confirmed by empirical studies (Vyas et al., 2023) and in (Lee et al., 2020; Seleznova and Kutyniok, 2022).

**Beyond the Lazy Regime** More recent papers (Damian et al., 2022; Lee et al., 2024; Mousavi-Hosseini et al., 2023) show that neural networks can achieve superior results to linearization or kernel methods. In particular, they can train high dimensional functions of type $g(Ux)$ with some intrinsic lower dimensionality given by a wide matrix $U \in \mathbb{R}^{r \times d}$ with $r \ll d$.

**Mean Field Limits** Earlier results (Chizat and Bach, 2018; Mei et al., 2018; Rotskoff and Vanden-Eijnden, 2018; Hu et al., 2020; Sirignano and Spiliopoulos, 2020; Bach and Chizat, 2021) consider the limiting objective (5) and particle discretizations. They show convergence to global minima for gradient flow time $t \to \infty$ and weight $m \to \infty$. Newer results contain finer grained quantitative error bounds for gradient flow time (Chizat, 2022; Nitanda et al., 2022) and discretization (Chen et al., 2023; Suzuki et al., 2023; Takakura and Suzuki, 2024).

### 1.5 NOTATIONS

The notations used throughout the paper are summarized in Appendix A.

## 2 CONTINUUM LIMIT AND APPROXIMATION

### 2.1 CONTINUUM LIMIT

Throughout this paper, we assume that the target function $f$ is given by

$$f = \int \sigma(w) \, d\phi(w), \tag{7}$$

where $\phi$ is contained in the space $\mathcal{M}$ of finite signed measures on $\mathcal{W}$. In case $\phi$ is not unique, we choose a minimizer of the Barron norm in (12), below.

Note that the continuum limit does not include outer weights $a$. While in mean field theory outer and inner weights $(a, w) \to v$ are often joined into one single variable, we marginalize instead. E.g. if $f$ is given by a probability density $\rho(a, w)$, we obtain

$$f = \iint \sigma(w) a \rho(a, w) \, da \, dw = \int \sigma(w) \left( \int a\rho(a, w) \, da \right) dw = \int \sigma(w) \, d\phi(w), \tag{8}$$

with signed measure $\phi(A) = \int_A \left( \int a\rho(a, w) \, da \right) dw$. Appendix C.2 shows that $\phi$ is often unique, unlike $\rho$, which can accommodate different distributions of $w$ by adapting the distribution of $a$.

### 2.2 APPROXIMATION: MAUREY SAMPLING

The main interest of continuum limits is to derive properties of their discrete practical counterparts. To study approximation properties, these are constructed by sampling.

In detail, we construct finite networks by replacing the integral representation (7) with a sample mean. However, we cannot sample from $\phi$ directly because it is signed and not normalized, hence not a probability. Therefore, let $\phi$ be absolutely continuous with respect to some probability measure $\pi$ from which we sample instead. Then, we can rewrite the continuum limit with the Radon-Nikodym derivative $a_\pi := d\phi/d\pi$ so that

$$f = \int \sigma(w) \frac{d\phi}{d\pi} \, d\pi(w) = \mathbb{E}_{w \sim \pi} \left[ a_\pi(w)\sigma(w) \right] \tag{9}$$

and replace the expectation by mean

$$f \approx \frac{1}{m} \sum_{i=1}^{m} a_\pi(w_i)\sigma(w_i), \tag{10}$$

with $\pi$-i.i.d samples $w_i$. To state error bounds, we define the *Barron norm relative to $\pi$*:

$$\|f\|_{B(\pi)} := \|f\| + |f|_{B(\pi)}, \qquad |f|^2_{B(\pi)} := \mathbb{E}_{w \sim \pi} \left[ \|a_\pi(w)\sigma(w)\|^2 \right]. \tag{11}$$

Then, along standard lines, (Weinan et al., 2022; Siegel and Xu, 2024) reviewed in Appendix C.1, we obtain:

**Theorem 2.1.** *Let $f$ and $\phi$ be given by (7). Let $\phi$ be absolutely continuous with respect to some probability measure $\pi$, let $a_\pi = d\phi/d\pi$ be the Radon-Nikodym derivative and let $w_i$, $i \in [m]$ be i.i.d sampled from $\pi$. Then*

$$\mathbb{E}\left\|\frac{1}{m}\sum_{i=1}^{m} a_\pi(w_i)\sigma(w_i) - f\right\|^2 \leq \frac{2}{m}\,|f|_{B(\pi)}^2\,.$$

We now turn to the best possible choice of $\pi$. Intuitively, we want to place the samples $w_i$ so that all summands in the sample mean (10) have the same weight. Formally, this means that the size of the summands $|a_\pi(w)|\,\|\sigma(w)\|$ is independent of $w$, which we refer to as *equidistribution*. The optimality of this choice is confirmed in Lemma 3.2 and leads to standard Barron type approximation error bounds (Weinan et al., 2022; Siegel and Xu, 2024), shown in Appendix C.1:

**Corollary 2.2.** *Let $f$ and $\phi$ be given by (7). For measurable sets $W \subset \mathcal{W}$, define the probability measure*

$$\pi(W) = |f|_B^{-1}\int_W \|\sigma(w)\|\,d|\phi|(w), \qquad\qquad |f|_B := \int_{\mathcal{W}} \|\sigma(w)\|\,d|\phi|(w).$$

*Let $a_\pi = d\phi/d\pi$ be the Radon-Nikodym derivative and $w_i$, $i \in [m]$ be i.i.d sampled from $\pi$. Then $|a_\pi(w)|\,\|\sigma(w)\| = |f|_{B(\pi)} = |f|_B$ $\pi$-almost surely and*

$$\mathbb{E}\left\|\frac{1}{m}\sum_{i=1}^{m} a_\pi(w_i)\sigma(w_i) - f\right\|^2 \leq \frac{2}{m}\,|f|_{B(\pi)}^2 = \frac{2}{m}\,|f|_B^2\,.$$

Based on the last corollary, we define the *Barron norm*

$$\|f\|_B = \|f\| + |f|_B\,, \qquad\qquad |f|_B = \int \|\sigma(w)\|\,d|\phi|(w). \qquad (12)$$

One usually takes the infimum over all eligible $\phi$, which we have already done in the definition (7). Our definition is similar to the Barron norms in Ongie et al. (2020), but there are several alternatives in the literature. E.g. the Barron norms in Weinan et al. (2022) use the continuum representation with outer weights as in (8) and define $|f|_B = \inf_\rho \mathbb{E}_{w\sim\pi}|a_\pi(w)|\|w\|$, which for ReLU activations is slightly larger than ours. An overview over alternative definitions of Barron norms is given in Siegel and Xu (2021).

The remarkable property of Theorem 2.1 and Corollary 2.2 is the dimension $d$ independent error rate $m^{-1/2}$, which can be improved slightly to $m^{-\frac{1}{2}-\frac{3}{2d}}$ for ReLU activations (Siegel and Xu, 2024). However, these results are of theoretical nature because practically we do not know $\phi$, $\pi$ or $a_\pi$. Also note that an optimal choice of the sampling distribution $\pi$ is paramount because the norm $\|f\|_{B(\pi)}$ can be significantly larger than the optimal $\|f\|_B$, even infinite, leading to poor performance of the discrete sample mean.

## 2.3 EXAMPLES

Appendix C.2 contains a simple model problem, where $\pi$ can be computed directly and equidistribution is comparable to classical approximation methods like splines or finite elements. The choice of sample distribution is also crucial in Monte Carlo integration $\int f(x)\,d\pi(x)$, where sampling from $\pi$ directly is preferred over sampling from generic distributions weighted by $\pi$'s density.

## 3 MAIN RESULTS

As we have seen in the last section, low approximation errors require a careful choice of the sample distribution $\pi$. Since all distributions, subject only to absolute continuity, achieve zero mean field loss (5), we use the more fine grained loss

$$\min_{\pi\in\mathcal{M}_{+,1}} \ell(\pi) := \min_{\pi\in\mathcal{M}_{+,1}} \mathbb{E}_{\boldsymbol{w}\sim\pi^m}\left\|\frac{1}{m}\sum_{i=1}^{m} a_\pi(w_i)\sigma(w_i) - f\right\|^2, \qquad (13)$$

which offers direct control over finite sized networks. Despite the product measure and nonlinear map $\pi \to a_\pi$, we show that local minimizers achieve equidistribution and Barron type approximation errors, first for idealized $a_\pi$ in Section 3.1 and then for more practical choices in Section 3.2.

## 3.1 EXACT OUTER WEIGHTS

In this section, for arbitrary probability measure $\phi \ll \pi \in \mathcal{M}_{+,1}$, we define the outer weights by the Radon-Nikodym derivative

$$a_\pi := \frac{d\phi}{d\pi} \qquad f = \int \sigma(w) \, d\phi(w) = \int \sigma(w) \frac{d\phi}{d\pi} \, d\pi(w) = \mathbb{E}_{w \sim \pi} \left[ a_\pi(w)\sigma(w) \right], \qquad (14)$$

where again we choose $\phi$ with minimal Barron norm $|f|_B$ in case it is not unique. Despite the product measure $\boldsymbol{w} \sim \pi^m$ and the dependence of $a_\pi$ on $\pi$, the loss $\ell(\pi)$ is convex on the set of permissible probability distributions, proven in Appendix E.1.

**Theorem 3.1.** *Let $\phi$ and $a_\pi$ be given by (14). Then the loss (13) is convex on the set of probability measures $\pi \in \mathcal{M}_{+,1}$ with $\phi \ll \pi$ and $a_\pi \in L^2(\pi)$.*

It follows that all local minima are global. The first order optimality criteria yield equidistribution, independent of $m$, shown in Appendix E.2.

**Lemma 3.2.** *Let $\pi$ be an absolutely continuous $\phi \ll \pi$ local minimizer of (13), with $a_\pi \in L^2(\pi)$ defined in (14) and $\|\sigma(\cdot)\| \in L^\infty(\pi)$. Then*

$$|a_\pi(w)| \, \|\sigma(w)\| = \lambda, \quad \pi\text{-almost surely,}$$

*for some $\lambda \in \mathbb{R}$.*

Corollary 2.2 suggests that $\lambda = |f|_B$, which justifies our definition of the Barron norm (12). We directly obtain the following approximation properties for finite networks sampled from optimizers $\pi$, proven in Appendix E.3.

**Theorem 3.3.** *Let $a_\pi \in L^2(\pi)$ given by (14), and $\pi$ be a local minimizer of the loss (13). Then*

$$\mathbb{E}_{\boldsymbol{w} \sim \pi^m} \left\| \frac{1}{m} \sum_{i=1}^m a_\pi(w_i)\sigma(w_i) - f \right\|^2 \leq \frac{2}{m} |f|_B^2 \, .$$

## 3.2 AVERAGED OUTER WEIGHTS

The results in the last section are based in the idealized outer weights $a_\pi = d\phi/d\pi$. They provide a streamlined theory, but are are not available in practice. Therefore, in this section we consider computable alternatives.

**Construction of Outer Weights: Best Approximation**  To construct practical approximations of $a_\pi$, we first consider finite sized neural networks: For width $M$, outer weights $\boldsymbol{a} \in \mathbb{R}^M$ and inner weights $\boldsymbol{w} \in \mathcal{W}^M$, define the shallow neural network

$$f_{\boldsymbol{a},\boldsymbol{w}} := \frac{1}{M} \sum_{i=1}^M a_i \sigma(w_i),$$

with optimal outer weights

$$\boldsymbol{a}(\boldsymbol{w}) := \operatorname*{arg\,min}_{\boldsymbol{a} \in \mathbb{R}^M}^+ \|f_{\boldsymbol{a},\boldsymbol{w}} - f\|^2 \, , \qquad\qquad f_{\boldsymbol{w}} := f_{\boldsymbol{a}(\boldsymbol{w}),\boldsymbol{w}}, \qquad (15)$$

where $\arg\min^+$ picks the candidate with minimal Euclidean norm, in case the minimizers are not unique. From these, we construct $\bar{a}_\pi(w)$, depending only on one single $w = w_1$ and distribution $\pi$, by averaging over all other weights

$$\bar{a}_\pi(w) := \mathbb{E}_{w_2,\ldots,w_M \sim \pi} \left[ \boldsymbol{a}_1(w, w_2, \ldots, w_M) \right].$$

We only pick the first component $\boldsymbol{a}_1$, which is multiplied with $\sigma(w) = \sigma(w_1)$ in the network. Unlike $a_\pi$ in the last section, the new $\bar{a}_\pi$ can be practically approximated by replacing the expectation by a sample mean.

The idealized coefficients $a_\pi$ are chosen so that the limiting network $\mathbb{E}_{w \sim \pi} \left[ a_\pi(w) \sigma(w) \right] = f$ matches the target function $f$ exactly. Lemma F.5 in the appendix shows that the new coefficients $\bar{a}_\pi$ satisfy this identity approximately

$$\mathbb{E}_{w \sim \pi} \left[ \bar{a}_\pi(w) \sigma(w) \right] = \mathbb{E}_{\boldsymbol{w} \sim \pi} \left[ f_{\boldsymbol{w}} \right],$$

where $f_{\boldsymbol{w}}$ is the best approximation of $f$ given inner weights $w_i$. We can ensure that $\mathbb{E}_{\boldsymbol{w} \sim \pi} \left[ f_{\boldsymbol{w}} \right]$ is close to $f$ by choosing sufficiently large width $M \gg m$ in the definition of $\bar{a}_\pi$, compared to width $m$ in the network loss (13).

**Construction of Outer Weights: Abstraction**    The main results in this section are still true if we replace the least squares minimizer by an arbitrary function $\boldsymbol{a} \colon \mathcal{W}^M \to \mathbb{R}^M$, subject only to the symmetry condition

$$P\boldsymbol{a}(\boldsymbol{w}) = \boldsymbol{a}(P\boldsymbol{w}), \qquad \text{for all permutation matrices } P \in \mathbb{R}^{M \times M}. \qquad (16)$$

This symmetry is satisfied by our original least squares coefficients (15) as shown in Lemma F.1. As before, we define

$$\bar{a}_\pi(w) := \mathbb{E}_{w_2, \dots, w_M \sim \pi} \left[ \boldsymbol{a}_1(w, w_2, \dots, w_M) \right], \qquad f_{\boldsymbol{w}} := f_{\boldsymbol{a}(\boldsymbol{w}), \boldsymbol{w}} \qquad (17)$$

and optimize the loss

$$\min_{\substack{\pi \in \mathcal{M}_+ \\ \pi(1)=1}} \ell(\pi) := \min_{\substack{\pi \in \mathcal{M}_+ \\ \pi(1)=1}} \mathbb{E}_{\boldsymbol{w} \sim \pi^m} \left\| \frac{1}{m} \sum_{i=1}^{m} \bar{a}_\pi(w_i) \sigma(w_i) - f \right\|^2. \qquad (18)$$

**Perturbation Terms**    The approximation $\bar{a}_\pi$ of $a_\pi$ results in extra perturbation terms $\Delta_1$ and $\Delta_2$, which we define next. With $g_\pi(w) = a_\pi(w)\sigma(w) - f$ and notations from Appendix A.2, set

$$\nu(\Delta_1) = 2 \sum_{j=1}^{M} \iint \left\langle g_\pi(w_j), \mathbb{E}_{\boldsymbol{w} \setminus 1, j} \left[ \boldsymbol{a}_j(\boldsymbol{w}) \right] \sigma(w_j) \right\rangle \, d\nu(w_1) \, d\pi(w_j), \qquad (19)$$

$$\nu(\Delta_1) = 2M \left\langle G_\pi, \nu \left( \mathbb{E}_{\boldsymbol{w} \setminus 1} \left[ f_{\boldsymbol{w}} - f_{\boldsymbol{w} \setminus 1} \right] \right) \right\rangle, \qquad (20)$$

$$\nu(\Delta_2) = 2(m-1)M \left\langle \mathbb{E}_{\boldsymbol{w}} \left[ f_{\boldsymbol{w}} - f \right], \nu \left( \mathbb{E}_{\boldsymbol{w} \setminus 1} \left[ f_{\boldsymbol{w}} - f_{\boldsymbol{w} \setminus 1} \right] \right) \right\rangle, \qquad (21)$$

for all $\nu \in \mathcal{M}$. For $\Delta_1$, we provide two variants that are equal up to an irrelevant constant by Lemma F.10 in the appendix: The first is elementary but more complicated. Note that for $j = 1$, the $w_1$ in the expectation is bound by the $\nu$ integral and therefore the outer $d\pi(w_1)$ integrates a constant and can be removed. The second variant is a simplification given existence of some $G_\pi \in \mathcal{H}$, independent of $w$ such that

$$\left\langle \bar{a}_\pi(w)\sigma(w) - f, \sigma(w) \right\rangle = \left\langle G_\pi, \sigma(w) \right\rangle, \qquad \pi - \text{a.s. in } w \in \mathcal{W}. \qquad (22)$$

The left hand side is a function of $w$ and for common choices of $\sigma$, by means of choosing $G_\pi$, the right hand side can represent arbitrary functions of $w$, subject to some regularity like Barron smoothness. E.g. for shallow neural networks without bias on the sphere $\mathcal{H} = L^2(\mathbb{S}^{d-1})$, the expression $\left\langle G_\pi, \sigma(w) \right\rangle = \int G_\pi(x) \operatorname{ReLU}(w^T x) \, dx$ is a function of $w$ in the same way the symmetric variant $f(x) = \int a(w) \operatorname{ReLU}(w^T x) \, dw$ is a function of $x$.

We postpone a more careful discussion of the perturbation terms to Appendix B. In short, we expect $|\Delta_1| \lesssim M^{-1/2}$ and $|\Delta_2| \lesssim mM^{-1/2}$ so that they vanish for $m \ll M$. Furthermore, if we choose $m = 1$, the second term is zero.

**Equidistribution**    Our first main result shows equidistribution analogous to Lemma 3.2, up to the two perturbation terms $\Delta_1$ and $\Delta_2$. The proof is in Appendix F.4.

**Lemma 3.4.** *Let $\pi$ be a local minimum of (18) with $\boldsymbol{a}_1 \in L^2(\pi^m)$, defined in (16), and perturbation terms $\Delta_1$ defined in (19) or (20) and $\Delta_2$ defined in (21). Then*

$$|\bar{a}_\pi(w)| \, \|\sigma(w)\| = \lambda + \Delta_1(w) + \Delta_2(w), \quad \pi\text{-almost surely}$$

*for some constant $\lambda \in \mathbb{R}$.*

**Approximation** In Section 2.2, we have seen how equidistribution leads to approximation results in Barron norms. We introduce a stable variant, to account for perturbation errors. For measure $\pi \in \mathcal{M}$, function $\delta \colon \mathcal{W} \to \mathbb{R}$, $\epsilon \geq 0$ and $\lambda \in \mathbb{R}$, consider the equidistribution bound

$$\left| \lambda - |a(w)|^2 \, \|\sigma(w)\|^2 \right| \leq \delta(w) \qquad\qquad \pi - \text{a.s.} \tag{23}$$

and the approximation bound

$$\|\mathbb{E}_{w \sim \pi}\left[a(w)\sigma(w)\right] - f\| \leq \epsilon. \tag{24}$$

Then we define the *stable Barron norm* by

$$|f|^2_{B(\delta,\epsilon)} := \sup \left\{ \lambda \in \mathbb{R} \big| \, \exists \pi \in \mathcal{M}_{+,1}, \exists a \in L^2(\pi) \text{ so that (23) and (24) are satisfied} \right\}.$$

To compare this with the Barron norm, we choose $\pi$ as in Corollary 2.2 and $a_\pi = d\phi/d\pi$. Then $\pi$ is a probability measure and the corollary shows that (23) and (24) are satisfied with perturbations $\delta = \epsilon = 0$ and $\lambda := |f|^2_B$. If the continuum limit is unique so that we can skip the minimum after (7), it follows that

$$|f|_{B(\delta=0,\epsilon=0)} = |f|_B.$$

The second main result shows approximation for stable Barron smoothness, analogous to Theorem 3.3. The proof is in Appendix F.5.

**Theorem 3.5.** *Let $\pi$ be a local minimum of (18) with $\boldsymbol{a}_1 \in L^2(\pi^m)$, defined in (16), and perturbation terms $\Delta_1$ defined in (19) or (20) and $\Delta_2$ defined in (21). Define*

$$\delta(w) := \bar{\lambda} + \Delta_1(w) + \Delta_2(w), \qquad\qquad \epsilon := \|\mathbb{E}_{\boldsymbol{w} \sim \pi}\left[f_{\boldsymbol{w}} - f\right]\|$$

*$\pi$-almost surely for arbitrary $\bar{\lambda} \in \mathbb{R}$. Then for all $\mathfrak{m} \in \mathbb{N}$*

$$\mathbb{E}_{\boldsymbol{w} \sim \pi^{\mathfrak{m}}} \left\| \frac{1}{\mathfrak{m}} \sum_{i=1}^{\mathfrak{m}} \bar{a}_\pi(w_i)\sigma(w_i) - f \right\|^2 \leq \frac{4}{\mathfrak{m}} \left[ |f|^2_{B(|\delta|,\epsilon)} + \pi(|\delta|) \right] + 2\epsilon^2.$$

Both Lemmas 3.2 and 3.4 show equidistribution $\pi$-almost surely. Hence, it may fail on sets of zero $\pi$-measure but non-zero Lebesgue measure. For the choice $a_\pi$, the absolute continuity $\phi \ll \pi$ ensures that such sets are irrelevant for the exact representation or approximation. However, the results for $\bar{a}_\pi$ have no such condition and $\pi$ may be zero on relevant subsets. However, such sets cannot be too important as long as the approximation error $\epsilon$ is small.

## 4 Towards practical algorithms

The main theorems provide equidistribution and approximation properties of locally optimal distributions $\pi$ of inner neural network weights. Practically, we need to sample or approximate these distributions. This section provides a preliminary and informal discussion along the lines of mean field theory.

**Wasserstein Gradient Flow** A natural candidate to optimize the losses (13) or (18) is Wasserstein gradient flow (WGF), given by

$$\dot{\pi} = \text{div}\left(\pi \nabla_w \nabla_\pi \ell(\pi)\right),$$

where $\nabla_\pi \ell(\pi)(w)$ denotes the Riesz representation $D[\ell(\pi)]\nu = \int \nabla_\pi \ell(\pi)(w)\,d\nu(w)$ of the directional derivatives. The WGF is implicitly understood in a distributional sense

$$\iint \left\{ \dot{\varphi}(t,w) - \nabla_w \varphi(t,w) \cdot \nabla_w \nabla_\pi \ell(\pi)(w) \right\} d\pi(w)\,dt = 0, \tag{25}$$

for all smooth and compactly supported functions $\varphi \colon [0,\infty) \times \mathcal{W} \to \mathbb{R}$.

To obtain a finite representation of $\pi$, we use a particle approximation, determined by weights $\boldsymbol{w} \in \mathbb{R}^{\mathfrak{m}}$ for large $\mathfrak{m} \gg m, M$, trained by regular gradient flow:

$$\pi_{\boldsymbol{w}} := \frac{1}{\mathfrak{m}} \sum_{i=1}^{\mathfrak{m}} \delta_{w_i}, \qquad\qquad \dot{\boldsymbol{w}} := -\mathfrak{m}\nabla_{\boldsymbol{w}}\ell(\pi_{\boldsymbol{w}}). \tag{26}$$

On the one hand, discretizing in time and expectation, leads to gradient descent methods with dropout type regularization, as we see in the next paragraph. On the other hand, gradient flow training of $\boldsymbol{w}$ matches WGF training of the corresponding distribution $\pi_{\boldsymbol{w}}$. The proof is well known (Chizat and Bach, 2018) and included for completeness in Appendix G.

**Lemma 4.1.** *Let the discrete measures $\pi_{\boldsymbol{w}(t)}$ be defined by gradient flow (26). Then they satisfy the Wasserstein gradient flow (25).*

The following result characterizes the stationary points of WGF, see Appendix G.

**Lemma 4.2.** *Let $\boldsymbol{w} \in \mathcal{W}^{\mathfrak{m}}$ be a stationary point of gradient flow, together with $\pi = \pi_{\boldsymbol{w}}$ defined in (26). Let $\bar{a}_\pi$ be defined by (17), with $\boldsymbol{a}_1 \in L^2(\pi)$, $\Delta_1$ be defined by (19) or (20) and $\Delta_2$ by (21). Then for all $i \in [\mathfrak{m}]$ we have*

$$\nabla_{w_i} \left[ -\mathfrak{m} \left\| \bar{a}_\pi(w_i)\sigma(w_i) \right\|^2 + \Delta_1(w_i) + \Delta_2(w_i) \right] = 0.$$

If $w$ was a continuous variable, the zero gradient would imply equidistribution on connected components of $\pi$'s support. Confined to discrete points, analogous conclusions require extra regularity, which is left for future work.

**Gradient Descent and Dropout**    To obtain a practical algorithm, we replace expectations with sample means

$$\bar{a}_\pi \approx \tilde{a}_\pi(v) = \frac{1}{N_a} \sum_{j=1}^{N_a} \boldsymbol{a}_1(v, \boldsymbol{v}_{\backslash 1}^j), \qquad\qquad \boldsymbol{v}^j \sim \pi^M, \qquad (27)$$

$$\ell(\pi) \approx \tilde{\ell}(\pi) = \frac{1}{N_\ell} \sum_{j=1}^{N_\ell} \left\| \frac{1}{m} \sum_{i=1}^{m} \tilde{a}_\pi(v_i^j)\sigma(v_i^j) - f \right\|^2, \qquad\qquad \boldsymbol{v}^j \sim \pi^m, \qquad (28)$$

for some numbers $N_a$ and $N_\ell$, and we replace gradient flow by gradient descent with learning rate $\gamma$

$$\boldsymbol{w}^{n+1} = \boldsymbol{w}^n - \gamma \nabla_{\boldsymbol{w}} \tilde{\ell}(\pi_{\boldsymbol{w}^n})$$

Since the samples $\boldsymbol{v}^j$ are a sub-selection of of the discrete $\boldsymbol{w}$, the gradient is well defined. Unravelling the discrete measure $\pi_{\boldsymbol{w}^n}$, this is a standard gradient descent iteration with two differences: First, instead of optimizing both $a_i$ and $w_i$ simultaneously, we ensure that the outer coefficients are always in an optimized state. Second, we apply two extra sample means. As we see in the following, these are comparable to dropout (Nitish et al., 2014) regularization in standard neural network training.

The connection is easiest for the outer expectation of $\ell(\pi)$. If we add all summands from the mean (28) as separate gradient descent steps and always use the newest possible sample distribution, we obtain the iteration

$$\boldsymbol{w}^{n+1} = \boldsymbol{w}^n - \frac{\gamma}{N_\ell} \nabla_{\boldsymbol{w}} \left\| \frac{1}{m} \sum_{i=1}^{m} \tilde{a}_\pi(v_i)\sigma(v_i) - f \right\|^2, \qquad\qquad \boldsymbol{v} \sim \pi_{\boldsymbol{w}^n}^m, \qquad (29)$$

see Appendix G.3 for details. Since $\pi_{\boldsymbol{w}}$ is a sum of Dirac deltas, the possible values of $v_i$ are given by the set $\{w_1, \ldots, w_{\mathfrak{m}}\}$. Hence, the sum over $m$ random elements of this set is identical to dropout of the full sum $\frac{1}{\mathfrak{m}} \sum_{i=1}^{\mathfrak{m}} \tilde{a}_{\pi_{\boldsymbol{w}}}(w_i)\sigma(w_i)$.

Similarly, intertwining gradient descent optimization and mean for the coefficients $\tilde{a}_\pi$ yields the iteration

$$\tilde{a}_\pi^n(v) = \tilde{a}_\pi^{n-1}(v) - \frac{\lambda}{N_a} \nabla_a \left\| \frac{1}{M} \sum_{i=1}^{M} \tilde{a}_\pi^{n-1}(v_i)\sigma(v_i) - f \right\|^2, \quad v_1 = v, \quad \boldsymbol{v}_{\backslash 1} \sim \pi^{M-1}, \quad (30)$$

for all particles $v \in \{w_1, \ldots, w_{\mathfrak{m}}\}$, see Appendix G.4 for details. Again the inner sum has random terms dropped out of the full sum over all $v_i \in \{w_1, \ldots, w_{\mathfrak{m}}\}$.

# 5 NUMERICAL EXPERIMENTS

In the following preliminary numerical experiments we train the function $f(x, y) = x^2$ on the domain $[-1, 1]^2$. While the domain is two dimensional, the function only depends on one variable to see if gradient descent can find this intrinsic low dimensionality. We train a shallow ReLU network on $n = 100$ uniformly random samples, with $500000$ gradient descent steps of learning rate $0.01$, widths $16, 32, 64, 128, 256$ and dropout $0, 0.5, 0.7$. Instead of computing $\bar{a}_\pi$, we include the outer weights in the regular gradient descent training.

Convergence rates are shown in Figure 1. Up to some outliers, they stay below the predicted $m^{-1/2}$. As rates in low dimensions can be higher (Siegel and Xu, 2024), this is not unexpected. Without dropout, the overall error is lower, because the outer coefficients are not disrupted by removed neurons.

Figure 2 shows $\|a_i \sigma(w_i)\|$ for all network indices $i$, sorted by magnitude. This quantity is equidistributed in the continuum limit in Lemmas 3.2 and 3.4 on the support of $\pi$. It seems that gradient descent either aligns the inner weights with the support or deactivates neurons by setting the outer coefficients to zero, so that we see two distinct plateaus in the figure. Note also that in Section 4 our loss leads to dropout regularization, without which the equidistribution disappears in the experiments.

The bottom row of Figure 1 shows the inner weights $v_i$ of $\sigma(w_i) = \text{ReLU}(v_i^T x + b_i)$ with $w_i = (v_i, b_i)$. Since the target $f$ is intrinsically one dimensional, all these points should be aligned with the $x$ axis. While this is not the case for unregularized training, dropout does achieve partial alignment. Interestingly, the aligned and nonaligned points belong to different plateaus in the equidistribution plots, as indicated by their color. Since this quantity is easily computable, this may be exploited for new neuron pruning and growing strategies in future work.

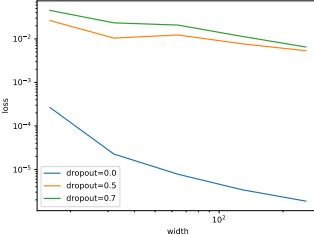

| m | dropout | | |
|---|---|---|---|
| | 0.0 | 0.5 | 0.7 |
| 16 | 3.56 | 1.34 | 0.95 |
| 32 | 1.54 | -0.24 | 0.17 |
| 64 | 1.18 | 0.67 | 0.86 |
| 128 | 0.87 | 0.53 | 0.81 |

Figure 1: Estimated convergence rates for varying width $m$ and dropout.

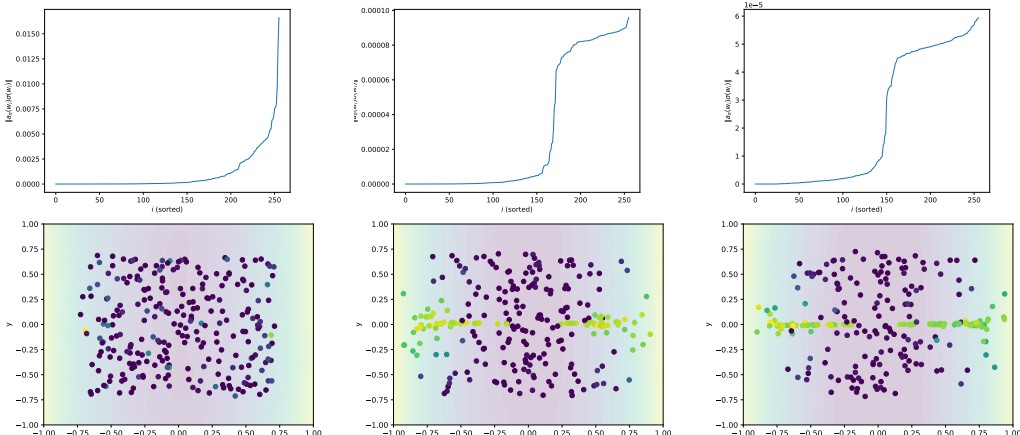

Figure 2: Left to right: Dropout $0.0$, $0.5$ and $0.7$. Top: Equidistribution (4) evaluated at the inner weights $w_i = (v_i, b_i)$. Bottom dots: inner weight $v_i$ of $\text{ReLU}(v_i^T x - b_i)$, colored by equidistribution (4). Bottom background: target function $f$.

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

## A NOTATIONS

### A.1 STANDARD NOTATIONS

For $m, M \in \mathbb{N}$, we define $[m] := \{1, \ldots, m\}$. For a vector $\boldsymbol{w} = [w_1, \ldots, w_M]$ and index set $\Lambda \subset [M]$, we write $\boldsymbol{w}_{\backslash \Lambda} := [w_i]_{i \in [M] \backslash \Lambda}$. For small sets we abbreviate $\boldsymbol{w}_{\backslash i} = \boldsymbol{w}_{\backslash \{i\}}$, $\boldsymbol{w}_{\backslash i,j} = \boldsymbol{w}_{\backslash \{i,j\}}$ etc. For $a_\pi \colon \mathcal{W} \to \mathbb{R}$, the expression $a_\pi(\boldsymbol{w})$, applied to a vector $\boldsymbol{w} \in \mathcal{W}^M$ is evaluated component-wise.

Throughout the paper $\mathcal{H}$ denotes a Hilbert space with norm $\|\cdot\|$ and inner product $\langle \cdot, \cdot \rangle$. Finite signed measures on $\mathcal{W}$ are denoted by $\mathcal{M}$, restriction to positive measures by $\mathcal{M}_+$ and probability measures by $\mathcal{M}_{+,1}$. For measures $\mu, \nu \in \mathcal{M}$, we denote the positive and negative parts by $\mu_+$ and $\mu_-$ and the variation by $|\mu|$. We abbreviate $\mu(f) := \int f(x)\, d\mu(x)$. We denote absolute continuity by $\mu \ll \nu$ and singularity by $\mu \perp \nu$. The product measure is $\mu^m := \bigotimes_{i=1}^m \mu$. We use standard $L^p(\pi)$ spaces for functions $\mathcal{W} \to \mathbb{R}$ and Bochner spaces $L^p(\pi; \mathcal{H})$ for functions $\mathcal{W} \to \mathcal{H}$. We denote the Gateaux derivative of $\ell(\pi)$ in direction $\nu$ by $D[\ell(\pi)]\nu$.

### A.2 FREE AND BOUND VARIABLES

We make judicious use of free and bound variables. For example in the expression

$$w \int_0^1 \sin(w)\, dw$$

the outer $w$ is a fee variable and can be replaced with any number. The inner $w$ is bound by the integral and can be renamed

$$w \int_0^1 \sin(v)\, dv$$

to avoid name collisions. The convention is typically used as follows: For some function $a \colon \mathbb{R}^M \to \mathbb{R}$, the expression $\mathbb{E}_{\boldsymbol{w}_{\backslash 1}}[a(\boldsymbol{w})]$ has bound variables $w_2, \ldots w_M$ and free variable $w_1$. This allows us to make sense of expressions like $\nu(\mathbb{E}_{\boldsymbol{w}_{\backslash 1}}[a(\boldsymbol{w})])$ for an arbitrary measure $\nu$. The outer $\nu(\cdot)$ denotes an integral with respect to $\nu$ and takes in a function. The inner expression is a function of the single free variable $w_1 \to \mathbb{E}_{\boldsymbol{w}_{\backslash 1}}[a(\boldsymbol{w})]$. We automatically bound the $\nu$ integration to the only free variable so that

$$\nu(\mathbb{E}_{\boldsymbol{w}_{\backslash 1}}[a(\boldsymbol{w})]) = \int \mathbb{E}_{w_2, \ldots, w_M}[a(w_1, w_2, \ldots, w_M)]\, d\nu(w_1).$$

This convention is analogous to free and bound variables in mathematical logic and after getting used to, it provides a compact notation that avoids a serious amount of indices and technicalities throughout this paper.

## B PERTURBATIONS

The main results Lemma 3.4 and Theorem 3.5 demonstrate equidistribution and approximation subject to two perturbation terms $\Delta_1$ and $\Delta_1$. In this section, we consider some preliminary simplifications and estimates.

For both $\Delta_1$ and $\Delta_2$, we need an estimate for $\mathbb{E}_{\boldsymbol{w}_{\backslash 1}}\left[\left\|f_{\boldsymbol{w}} - f_{\boldsymbol{w}_{\backslash 1}}\right\|\right]$. Standard approximation results from Theorem 2.1 yield

$$\mathbb{E}_{\boldsymbol{w}_{\backslash 1}}\left[\left\|f_{\boldsymbol{w}} - f_{\boldsymbol{w}_{\backslash 1}}\right\|\right] \leq \mathbb{E}_{\boldsymbol{w}_{\backslash 1}}\left[\|f_{\boldsymbol{w}} - f\|\right] + \mathbb{E}_{\boldsymbol{w}_{\backslash 1}}\left[\left\|f - f_{\boldsymbol{w}_{\backslash 1}}\right\|\right] = \mathcal{O}(M^{-1/2}),$$

which is insufficient because both $\Delta_1$ and $\Delta_2$ contain a factor of $M$. However, the estimate is rather crude since usually $f_{\boldsymbol{w}_{\backslash 1}}$ is much closer to $f_{\boldsymbol{w}}$ than $f$, which is exploited in the following lemma and yields an improved error of $\mathcal{O}(M^{-1})$. The proof is in Appendix H.2.

**Lemma B.1.** *Let $\boldsymbol{a}$ and $f_{\boldsymbol{w}}$ be given by* (15). *Then, we have*

$$\mathbb{E}_{\boldsymbol{w}_{\backslash 1}}\left\|f_{\boldsymbol{w}} - f_{\boldsymbol{w}_{\backslash 1}}\right\| \leq M^{-1}\mathbb{E}_{\boldsymbol{w}_{\backslash 1}}[|\boldsymbol{a}_1(\boldsymbol{w})|]\, \|\sigma(w_1)\|.$$

The main obstacle to proceed further is the absolute value around $\boldsymbol{a}_1$. Since $f_{\boldsymbol{w}} \to f$ for $M \to \infty$, one may expect that also $\boldsymbol{a}_1(\boldsymbol{w}) \to a_\pi(w_1)$. Then, in the limit coefficients are non-oscillatory in the sense that $\mathbb{E}_{\boldsymbol{w}_{\backslash 1}}\left[|a(\boldsymbol{w})_1|\right] \lesssim |\mathbb{E}_{\boldsymbol{w}_{\backslash 1}}\left[a(\boldsymbol{w})_1\right]| = |\bar{a}_\pi(w_1)|$ and the right hand side is roughly the Barron norm $M^{-1}|f|_B$. This allows us to further bound the perturbation terms as in the following corollary. To state the result, we use the space

$$\mathcal{H}(\boldsymbol{w}) := \text{span}\{\sigma(w_i) : i \in [M]\} \tag{31}$$

and likewise for shortened vectors like $\boldsymbol{w}_{\backslash i}$. For the proof, see Appendix H.3.

**Corollary B.2.** *Let $\pi$ be a local minimum of (18) with $G_\pi$ defined in (22), best approximations $\boldsymbol{a}_1 \in L^2(\pi)$, $f_{\boldsymbol{w}}$ defined in (15) and $\mathcal{H}(\boldsymbol{w})$ defined in (31). Assume*

$$\mathbb{E}_{\boldsymbol{w}_{\backslash 1}}\left[|\boldsymbol{a}_1(\boldsymbol{w})|\right] \leq c|\mathbb{E}_{\boldsymbol{w}_{\backslash 1}}\left[\boldsymbol{a}_1(\boldsymbol{w})\right]|$$

*for some $c \geq 0$ and define*

$$\Delta := 2c\Big[\inf_{h \in \mathcal{H}(\boldsymbol{w}_{\backslash 1})} \|G_\pi - h\| + (m-1)\|\mathbb{E}_{\boldsymbol{w}}\left[f_{\boldsymbol{w}} - f\right]\|\Big]\|\bar{a}_\pi\sigma\|_{L^\infty(\pi;\mathcal{H})},$$

$$\epsilon := \|\mathbb{E}_{\boldsymbol{w}\sim\pi}\left[f_{\boldsymbol{w}} - f\right]\|.$$

*Then for all $\mathfrak{m} \in \mathbb{N}$*

$$\mathbb{E}_{\boldsymbol{w}\sim\pi^{\mathfrak{m}}}\left\|\frac{1}{\mathfrak{m}}\sum_{i=1}^{\mathfrak{m}}\bar{a}_\pi(w_i)\sigma(w_i) - f\right\|^2 \leq \frac{4}{\mathfrak{m}}\left[|f|^2_{B(\Delta,\epsilon)} + \pi(\Delta)\right] + \epsilon^2.$$

We may expect that $\|\bar{a}_\pi\sigma\|_{L^\infty(\pi;\mathcal{H})}$ is bounded, that $\|\mathbb{E}_{\boldsymbol{w}}\left[f_{\boldsymbol{w}} - f\right]\| \lesssim M^{-1/2}$ and, similar to approximation by neural networks, that $\inf_{h \in \mathcal{H}(\boldsymbol{w}_{\backslash 1})}\|G_\pi - h\| \lesssim M^{-1/2}$. Inspecting the proof, this yields the perturbations $\Delta = |\Delta_1 + \Delta_2|$ with $|\Delta_1| \lesssim M^{-1/2}$, $|\Delta_2| \lesssim mM^{-1/2}$. In conclusion, for non-oscillatory coefficients, and $M$ sufficiently larger than $m$, the perturbation terms are negligible and we obtain Barron type approximation results.

## C   MAUREY SAMPLING AND BARRON SPACES

### C.1   PROOF OF THEOREM 2.1 AND COROLLARY 2.2

In this section, we prove Theorem 2.1 and Corollary 2.2. These are minor adaptions from Theorem 4 in Weinan et al. (2022) and Theorem 1 in Siegel and Xu (2024).

**Lemma C.1.** *Let $X_i \in \mathcal{H}$ be mean zero independent random variables. Then for independent copies $X_i'$ of $X_i$*

$$\mathbb{E}\left\|\frac{1}{m}\sum_{i=1}^{m}X_i\right\|^2 = \frac{1}{2m^2}\sum_{i=1}^{m}\mathbb{E}\mathbb{E}'\|X_i - X_i'\|^2 = \frac{1}{m^2}\sum_{i=1}^{m}\mathbb{E}\|X_i\|^2$$

*Proof.* For the first and last term we have

$$\mathbb{E}\left\|\frac{1}{m}\sum_{i=1}^{m}X_i\right\|^2 = \frac{1}{m^2}\sum_{i=1}^{m}\sum_{i=1}^{m}\mathbb{E}\langle X_i, X_j\rangle = \frac{1}{m^2}\sum_{i=1}^{m}\mathbb{E}\|X_i\|^2.$$

For the middle term we have

$$\frac{1}{2m^2}\sum_{i=1}^{m}\mathbb{E}\mathbb{E}'\|X_i - X_i'\|^2 = \frac{1}{2m^2}\sum_{i=1}^{m}\mathbb{E}\mathbb{E}'\left[\|X_i\|^2 + \|X_i'\|^2 - 2\langle X_i, X_i'\rangle\right] = \frac{1}{m^2}\sum_{i=1}^{m}\mathbb{E}\|X_i\|^2,$$

which concludes the proof.

$\square$

*Proof of Theorem 2.1.* Define the random variables $X_i := a_\pi(w_i)\sigma(w_i) - f$. Then by construction (9) the random variables have zero mean $\mathbb{E}[X_i] = 0$, are i.i.d. and by Lemma C.1 we have

$$\mathbb{E}\left\|\frac{1}{m}\sum_{i=1}^m X_i\right\|^2 \le \frac{1}{2m^2}\mathbb{E}\mathbb{E}'\sum_{i=1}^m \|X_i - X_i'\|^2 \le \frac{2}{m^2}\mathbb{E}\sum_{i=1}^m \|a_\pi(w_i)\sigma(w_i)\|^2 = \frac{2}{m}|f|_{B(\pi)}^2,$$

where the last equality follows from the definition (11) of the Barron norm.

$\square$

*Proof of Corollary 2.2.* First note that the normalization factor $|f|_B^{-1}$ ensures that $\pi(\mathcal{W}) = 1$ so that $\pi$ is a probability measure. Since for all continuous functions $f$ by the definition of $\pi$ we have

$$\int f(w)\,d|\phi|(w) = \int f(w)\frac{|f|_B}{\|\sigma(w)\|}\frac{\|\sigma(w)\|}{|f|_B}\,d|\phi|(w) = \int f(w)\frac{|f|_B}{\|\sigma(w)\|}\,d\pi(w),$$

we must have $\frac{d|\phi|}{d\pi} = \frac{|f|_B}{\|\sigma(w)\|}$, almost surely and thus $|a_\pi| = \left|\frac{d\phi}{d\pi}\right| = \frac{d|\phi|}{d\pi} = \frac{|f|_B}{\|\sigma(w)\|}$. It follows that $|a_\pi(w)|\,\|\sigma(w)\| = |f|_B$ almost surely, which directly implies $|f|_{B(\pi)} = |f|_B$ and the stated error bound by Theorem 2.1.

$\square$

## C.2 EXAMPLE

To better understand the role of $\pi$, it is instructive to consider the approximation of a target function $f\colon [0,1] \to \mathbb{R}$ in one dimension with $\sigma(w) := \mathrm{ReLU}(x - w)$. In this setup, shallow networks of width $m$ are identical to much better understood linear splines with $m$ free knots at locations $w_i$. We would like to place most knots, where $f$ is complicated, measured by high derivatives, e.g. for $f(x) = \sqrt{x}$ as follows:

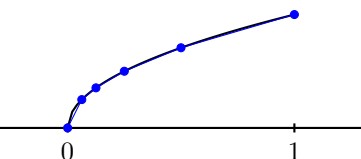

Such an adaptive placement of knots is achieved in the construction of Barron approximation results in Corollary 2.2, which can be explicitly computed as follows. For $\phi$ given by density $d\phi(w) = \rho_\phi(w)\,dw$ so that $f(x) = \int \mathrm{ReLU}(x - w)\rho_\phi(w)\,dw$, differentiating twice, yields

$$f''(x) = \int \delta(x - w)\rho_\phi(w)\,dw = \rho_\phi(x)$$

and thus the density $\rho_\phi = f''$ is the second derivative. Then the sample distribution $\pi$ is given by density $\rho_\pi(w) \sim \|\sigma(w)\|\,|\rho_\phi(w)| \sim \|\sigma(w)\|\,|f''(w)|$. Note carefully that regions where $f$ is difficult to approximate, indicated by a large derivative $|f''(w)|$, have equally large density. Hence, sampling places most knots $w_i$ in difficult regions as desired.

For other sample distributions $\pi$, the approximation results can be worse or even fail. Indeed, the Barron norm with respect to arbitrary $\pi$ is

$$\|f\|_{B(\pi)} = \mathbb{E}\|a_\pi(w)\sigma(w)\|^2, \qquad a_\pi(w) = \frac{d\phi}{d\pi}(w) = \frac{f''(w)}{\rho_\pi(w)},$$

which can be infinite, even if the optimal choice $\|f\|_B$ is finite. E.g. for uniform $\pi$ and $f(x) = |x - 1/2|^\alpha$ with $\alpha \in (1.0, 1.5)$. In this case, Theorem 2.1 and Corollary 2.2 provide approximation rate $m^{-1/2}$ for sampling from the optimal distribution but none for sampling from the uniform distribution.

The argument for this simple model problem relies in the observation that the second derivative of a ReLU activation is a Dirac delta. This remains true for standard shallow networks in multiple

dimensions if we replace the second derivative with higher derivatives, a ramp filter and a Radon transform, as shown in Parhi and Nowak (2021, Lemma 17). The problem of finding a good sample distribution $\pi$ is even more important in high dimensions as it must reveal intrinsic low dimensional structures of the target $f$.

# D  AUXILIARY RESULTS

## D.1  GRADIENTS

This section provides several basic gradients that are used repeatedly in the proofs of the main results. Recall that $\mathcal{M}$ denotes the vector space of finite signed measures on $\mathcal{W}$ and $\mathcal{H}$ denotes a Hilbert space.

**Lemma D.1.** *For $\pi, \nu \in \mathcal{M}$, let $g \in L^2(|\pi|; \mathcal{H}) \cap L^2(|\nu|; \mathcal{H})$. Then the flowing two Gateaux derivatives are well defined:*

$$D_\pi \left[ \int \|g(w)\|^2 \, d\pi(w) \right] \nu = \int \|g(w)\|^2 \, d\nu(w),$$

$$D_\pi \left[ \int \langle g(v), g(w) \rangle \, d\pi^2(v, w) \right] \nu = 2 \int \langle g(v), g(w) \rangle \, d(\pi \otimes \nu)(v, w).$$

*Proof.* Follows directly from elementary computation of the difference quotients. All necessary integrals are finite by the given assumptions. $\qquad\square$

**Lemma D.2.** *For $\pi, \nu \in \mathcal{M}$, let $g \in L^2(|\pi|; \mathcal{H}) \cap L^2(|\nu|; \mathcal{H})$. Then the flowing Gateaux derivative is well defined:*

$$D_\pi \left[ \int \left\| \sum_{i=1}^m g(w_i) \right\|^2 \, d\pi^n(\boldsymbol{w}) \right] \nu$$
$$= m\pi(1)^{m-1} \nu\left( \|g\|^2 \right) + 2m(m-1)\pi(1)^{m-2} \langle \pi(g), \nu(g) \rangle + \lambda\nu(1)$$

*for some constant $\lambda \in \mathbb{R}$.*

*Proof.* Let $D$ be the derivative in the lemma. It follows from Lemma D.1 that the derivative exists with

$$D = \sum_{k=1}^m \int \left\| \sum_{i=1}^m g(w_i) \right\|^2 \prod_{\substack{j=1 \\ j\neq k}}^m d\pi(w_j) \, d\nu(w_k).$$

By symmetry, we can permute each summand so that $k$ is replaced by 1 and the expression simplifies to

$$D = m \iint \left\| \sum_{i=1}^m g(w_i) \right\|^2 \prod_{j=2}^m d\pi(w_j) \, d\nu(w_1) = m(\nu \otimes \pi^{m-1}) \left( \left\| \sum_{i=1}^m g(w_i) \right\|^2 \right).$$

We abbreviate $G(\boldsymbol{w}_{\backslash 1}) := \sum_{i=2}^m g(w_i)$ and split the norm accordingly

$$D = m(\nu \otimes \pi^{m-1}) \left( \|g(w_1)\|^2 \right)$$
$$+ 2m(\nu \otimes \pi^{m-1}) \left( \langle g(w_1), G(\boldsymbol{w}_{\backslash 1}) \rangle \right)$$
$$+ m(\nu \otimes \pi^{m-1}) \left( \|G(\boldsymbol{w}_{\backslash 1})\|^2 \right)$$
$$=: I + II + III.$$

The first summand simplifies to

$$I = m\nu\left( \|g\|^2 \right) \pi^{m-1}(1)$$

and the second to

$$II = 2m \left\langle \nu(g), \pi^{m-1}(G) \right\rangle = 2m(m-1) \left\langle \nu(g), \pi(g) \right\rangle \pi^{m-2}(1),$$

where in the last step we have used that $\pi^{m-1}(G) = \sum_{i=2}^{m} \pi(g)\pi^{m-2}(1) = (m-1)\pi(g)\pi^{m-2}(1)$. Finally, the last term is

$$III = m\nu(1)\pi^{m-1}\left( \left\| G(\boldsymbol{w}_{\backslash 1}) \right\|^2 \right) = \lambda \nu(1),$$

for some constant $\lambda \in \mathbb{R}$ independent of $\nu$. Combining all summands shows the lemma.

$\square$

**Lemma D.3.** *For $\pi, \nu \in \mathcal{M}$, let $g_\pi \in L^2(|\pi|, \mathcal{H})$ be Gateaux differentiable with respect to $\pi \in \mathcal{M}$ and direction $\nu \in \mathcal{M}$ so that $w \to D_\pi[g_\pi(w)]\nu \in L^2(|\pi|, \mathcal{H})$. Then the flowing Gateaux derivative is well defined:*

$$\int D_\pi \left[ \left\| \sum_{i=1}^{m} g_\pi(w_i) \right\|^2 \right] \nu \, d\pi^n(\boldsymbol{w})$$

$$= m\pi(1)^{m-1} \int D_\pi \left[ \|g_\pi(w)\|^2 \right] \nu \, d\pi(w) + 2m(m-1)\pi(1)^{m-2} \left\langle \pi(g_\pi), \pi(D_\pi[g_\pi]\nu) \right\rangle.$$

*Proof.* Since the outer function $\|\cdot\|^2 : \mathcal{H} \to \mathbb{R}$ is Fréchet differentiable and the inner functions $\pi \to g_\pi(w_i)$ are Gateaux differentiable, by the chain rule we have

$$D_\pi \left[ \left\| \sum_{i=1}^{m} g_\pi(w_i) \right\|^2 \right] \nu = \sum_{\substack{i=1 \\ j=1}}^{m} 2 \left\langle g_\pi(w_i), D_\pi[g_\pi(w_j)]\nu \right\rangle =: \sum_{\substack{i=1 \\ j=1}}^{m} D_{ij}.$$

Undoing the derivative for the $i = j$ terms, we obtain

$$D_{ii} = D_\pi \left[ \|g_\pi(w_i)\|^2 \right] \nu$$

and thus by integrating and renaming the integration variable $w_i \to w$

$$\pi^m(D_{ii}) = \int D_\pi \left[ \|g_\pi(w)\|^2 \right] \nu \, d\pi(w) \pi(1)^{m-1}.$$

For $i \neq j$, the variables $w_i$ and $w_j$ in the inner product are independent so that

$$\pi^m(D_{ij}) = 2 \left\langle \pi(g_\pi), \pi(D_\pi[g_\pi]\nu) \right\rangle \pi(1)^{m-2}.$$

Noting that the last two right hand sides are independent of $i$ and $j$, we can eliminate the $i, j$ sum above and obtain the statement of the lemma.

$\square$

**Lemma D.4.** *Let $\pi, \nu \in \mathcal{M}$ with $\pi(1) = 1$ and $\|\sigma(\cdot)\| \in L^\infty(|\pi|)$. Let $g_\pi(w) = a_\pi(w)\sigma(w) - f$ for some $a_\pi \in L^2(|\pi|; \mathcal{H}) \cap L^2(|\nu|; \mathcal{H})$, Gateaux differentiable with respect to $\pi \in \mathcal{M}$ and direction $\nu \in \mathcal{M}$ so that $w \to D_\pi[a_\pi(w)]\nu \in L^2(|\pi|) \cap L^2(|\nu|)$. Let $\pi, \nu$, both be absolutely continuous with respect to some measure $\mu \in \mathcal{M}$. Then there is a $\lambda \in \mathbb{R}$ with*

$$D_\pi \left[ \int \left\| \sum_{i=1}^{m} g_\pi(w_i) \right\|^2 d\pi^n(\boldsymbol{w}) \right] \nu = I + II + III + IV + V,$$

*with*

$$I = -m \int \|a_\pi(w)\sigma(w)\|^2 \, d\nu(w)$$

$$II = 2m \int \left\langle g_\pi(w), D_\pi \left[ a_\pi(w)\sigma(w)\frac{d\pi}{d\mu} \right] \nu \right\rangle d\mu(w)$$

$$III = 2m(m-1) \left\langle \pi(g_\pi), \nu(g_\pi) \right\rangle$$

$$IV = 2m(m-1) \left\langle \pi(g_\pi), \pi(D_\pi[g_\pi]\nu) \right\rangle$$

$$V = \lambda \nu(1).$$

*for some $\lambda \in \mathbb{R}$.*

*Proof.* Let $D$ be the derivative in the lemma. The two terms dependent on the argument $\pi$ are $g_\pi$ and $d\pi^n(\boldsymbol{w})$ so that, by the product rule we have

$$D = \int D_\pi \left[ \left\| \sum_{i=1}^m g_\pi(w_i) \right\|^2 \right] \nu \, d\pi^n(\boldsymbol{w}) + D_\pi \left[ \int \left\| \sum_{i=1}^m g_\mu(w_i) \right\|^2 d\pi^n(\boldsymbol{w}) \right] \nu \Bigg|_{\mu=\pi},$$

where in the second term the inner $g_\pi$ is replaced by $g_\mu$ for some temporary $\mu = \pi$ so that it is not differentiated. The derivatives in the right hand side are given by Lemmas D.3 and D.2. Together with the normalization $\pi(1) = 1$, they yield

$$D = (i + IV) + (ii + III + V),$$

with

$$i = m \int D_\pi \left[ \| g_\pi(w) \|^2 \right] \nu \, d\pi(w),$$

$$IV = 2m(m-1) \left\langle \pi(g_\pi), \pi(D_\pi[g_\pi]\nu) \right\rangle,$$

$$ii = m\nu \left( \| g_\pi \|^2 \right),$$

$$III = 2m(m-1) \left\langle \pi(g_\pi), \nu(g_\pi) \right\rangle,$$

$$V = \lambda \nu(1).$$

The terms $III$, $IV$ and $V$ match the corresponding terms in the conclusion of the lemma. The terms $i$ and $ii$ require some attention. For the former, using the chain rule and $D_\pi f = 0$, we obtain

$$i = 2m \int \left\langle g_\pi(w), D_\pi[g_\pi(w)]\nu \right\rangle \, d\pi(w) = 2m \int \left\langle g_\pi(w), D_\pi\left[ a_\pi(w)\sigma(w) \right]\nu \right\rangle \, d\pi(w).$$

Since $\pi$ is absolutely continuous with respect to $\mu$, there is a Radon-Nikodym derivative $d\pi = \frac{d\pi}{d\mu} d\mu$ so that

$$i = 2m \int \left\langle g_\pi(w), D_\pi\left[ a_\pi(w)\sigma(w) \right]\nu \right\rangle \frac{d\pi}{d\mu} \, d\mu(w).$$

Then, with $D_\pi \left[ \frac{d\pi}{d\mu} \right] \nu \, d\mu = \frac{d\nu}{d\mu} d\mu = d\nu$ the product rule entails

$$i = 2m \int \left\langle g_\pi(w), D_\pi\left[ a_\pi(w)\sigma(w)\frac{d\pi}{d\mu} \right]\nu \right\rangle \, d\mu(w)$$

$$- 2m \int \left\langle g_\pi(w), a_\pi(w)\sigma(w) \right\rangle \, d\nu(w)$$

$$=: II + i.ii.$$

Finally, invoking the definition of $g_\pi = a_\pi\sigma - f$, we conclude that

$$i.ii = -2m \int \| a_\pi(w)\sigma(w) \|^2 \, d\nu(w) + 2m \int \left\langle f, a_\pi(w)\sigma(w) \right\rangle \, d\nu(w).$$

Likewise, the term $ii$ expands to

$$ii = m \int \| a_\pi(w)\sigma(w) \|^2 \, d\nu(w) - 2m \int \left\langle f, a_\pi(w)\sigma(w) \right\rangle \, d\nu(w) + m \int \| f \|^2 \, d\nu(w).$$

Combining $i.ii$ and $ii$ and cancelling terms, we obtain

$$i.ii + ii = -m \int \| a_\pi(w)\sigma(w) \|^2 \, d\nu(w) + m \int \| f \|^2 \, d\nu(w) =: I + V.i.$$

Joining $V + V.i$ by adjusting the constant $\lambda \to \lambda + m \| f \|^2$, and adding all terms proves the lemma.

$\square$

## D.2 FIRST ORDER OPTIMALITY CRITERIA

In this section, we briefly review first order optimality criteria for the optimization of probability measures. To this end, we optimize a function $\ell \colon \mathcal{M} \to \mathbb{R}$, confined to positive and normalized measures:

$$\min_{\pi \in \mathcal{M}_{+,1}} \ell(\pi).$$

For positive measures $\nu \in \mathcal{M}_+$, by a short computation and rescaling, the one-sided first order optimality criteria for the convex combinations $t \to \ell(\pi + t(\nu - \pi))$ at $t = 0$ yield

$$D\ell(\pi)(\nu) + \lambda\nu(1) \geq 0, \qquad\qquad D\ell(\pi)(\pi) + \lambda\pi(1) = 0, \qquad (32)$$

for some constant $\lambda \in \mathbb{R}$. With the definition $-h(\nu) := D\ell(\pi)(\nu) + \lambda\nu(1)$, we can extend the criteria to arbitrary $\nu \in \mathcal{M}$ and obtain standard KKT conditions:

$$\begin{aligned}
\text{stationarity:} &\quad D\ell(\pi)(\nu) + \lambda\nu(1) + h(\nu) = 0, &\quad \nu \in \mathcal{M} \\
\text{dual feasibility:} &\quad h(\nu) \leq 0, &\quad \nu \in \mathcal{M}_+ \\
\text{complementary slackness:} &\quad h(\pi) = 0.
\end{aligned}$$

# E  PROOF OF MAIN RESULTS: EXACT OUTER WEIGHTS

## E.1 PROOF OF THEOREM 3.1: CONVEXITY

To show convexity of the loss

$$\ell(\pi) := \mathbb{E}_{\boldsymbol{w} \sim \pi^m} \left\| \frac{1}{m} \sum_{i=1}^m a_\pi(w_i)\sigma(w_i) - f \right\|^2,$$

we first expand the squared norm

$$\ell(\pi) = \mathbb{E}_{\boldsymbol{w} \sim \pi^m} \left\| \frac{1}{m} \sum_{i=1}^m a_\pi(w_i)\sigma(w_i) \right\|^2 - 2\mathbb{E}_{\boldsymbol{w} \sim \pi^m} \left\langle \frac{1}{m} \sum_{i=1}^m a_\pi(w_i)\sigma(w_i), f \right\rangle + \|f\|^2.$$

$$=: I + II + III.$$

The third term $III$ is constant and therefore convex. For the second term $II$, we plug in the definition of $a_\pi = d\phi/d\pi$ to obtain

$$II = -2\frac{1}{m} \sum_{i=1}^m \mathbb{E}_{\boldsymbol{w}_{\backslash i}} \int \langle \sigma(w_i), f \rangle \frac{d\phi}{d\pi}(w_i) \, d\pi(w_i)$$

$$= -2\frac{1}{m} \sum_{i=1}^m \int \langle \sigma(w_i), f \rangle \, d\phi(w_i),$$

where we have dropped the expectation $\mathbb{E}_{\boldsymbol{w}_{\backslash i}}$ because its argument is independent of $\boldsymbol{w}_{\backslash 1}$. This term is independent of $\pi$ and thus convex.

It remains to show that the first term is also convex. To this end, we first expand the inner sum and split indices $i = j$ and $i \neq j$

$$I = \frac{1}{m^2} \sum_{i=1}^m \mathbb{E}_{\boldsymbol{w} \sim \pi^m} \|a_\pi(w_i)\sigma(w_i)\|^2 + \frac{1}{m^2} \sum_{i \neq j} \mathbb{E}_{\boldsymbol{w} \sim \pi^m} \langle a_\pi(w_i)\sigma(w_i), a_\pi(w_j)\sigma(w_j) \rangle$$

$$=: I.1 + I.2.$$

The term $I.2$ is analogous to $II$: We plug in the definition of $a_\pi$, and remove unnecessary expectations $\mathbb{E}_{w_k}$ for $k \neq i, j$ to obtain

$$I.2 = \frac{1}{m^2} \sum_{i \neq j} \iint \langle \sigma(w_i), \sigma(w_j) \rangle \frac{d\phi}{d\pi}(w_i) \frac{d\phi}{d\pi}(w_j) \, d\pi(w_i) \, d\pi(w_j)$$

$$= \frac{1}{m^2} \sum_{i \neq j} \iint \langle \sigma(w_i), \sigma(w_j) \rangle \, d\phi(w_i) \, d\phi(w_j),$$

which is independent of $\pi$ and therefore convex. Noting that all expectations in $I.1$ are identical, we simplify to

$$mI.1 = \int \|\sigma(w)\|^2 \left|\frac{d\phi}{d\pi}\right|^2 (w) \, d\pi(w)$$

$$= \int \|\sigma(w)\|^2 \left(\frac{d|\phi|}{d\pi}\right)^2 (w) \, d\pi(w) = \int \|\sigma(w)\|^2 \frac{d|\phi|}{d\pi}(w) \, d|\phi|(w).$$

By the Lebesgue decomposition theorem, we can split $\pi = \pi_0 + \pi_1$ into a part that is absolutely continuous $\pi_0 \ll |\phi|$ and a part that is singular $\pi_1 \perp |\phi|$ with respect to $|\phi|$. Since by assumption $\phi_\pm \ll \phi \ll \pi$, it is not hard to see that $\pi_0 \ll |\phi| \ll \pi_0$ and $d|\phi|/d\pi = d|\phi|/d\pi_0 = (d\pi_0/d|\phi|)^{-1}$ on the support of $|\phi|$, so that

$$mI.1 = \int \|\sigma(w)\|^2 \left(\frac{d\pi_0}{d|\phi|}\right)^{-1} (w) \, d|\phi|(w).$$

The functions $\pi \to \pi_0$ and $\pi_0 \to d\pi_0/d|\phi|$ are linear and since $\pi_0$ and $|\phi|$ are non-negative measures, $d\pi_0/d|\phi|$ is non-negative, as well, so that the inverse $x \to x^{-1}$ is convex. Since all remaining terms in the integral of $I.1$ are also non-negative, it follows that $I.1$ is convex in $\pi$. Together with the convexity of $I.2$, $II$ and $III$, this completes the proof.

### E.2 PROOF OF LEMMA 3.2: EQUIDISTRIBUTION

We first compute the derivative of the loss

**Lemma E.1.** *Let $\nu \ll \pi \in \mathcal{M}$ with $\pi(1) = 1$ and $\|\sigma(\cdot)\| \in L^\infty(|\pi|)$, and bounded Radon-Nikodym derivative $d\nu/d\pi$. Let $a_\pi \in L^2(|\pi|)$ be given by (14). Then with $g_\pi(w) = a_\pi(w)\sigma(w) - f$, there is a $\lambda \in \mathbb{R}$ so that*

$$D_\pi \left[ \int \left\| \sum_{i=1}^m g_\pi(w_i) \right\|^2 d\pi^n(\boldsymbol{w}) \right] \nu = -m \int \|a_\pi(w)\sigma(w)\|^2 \, d\nu(w) + \nu(\lambda),$$

*Proof.* Let $D$ be the derivative in the lemma, which we compute by Lemma D.4. To this end, we first establish its assumptions. Since $d\nu/d\pi$ is bounded, we have

$$\int |a_\pi(w)|^2 d\nu(w) = \int |a_\pi(w)|^2 \frac{d\nu}{d\pi}(w) d\pi(w) \lesssim \int |a_\pi(w)|^2 d\pi(w) < \infty$$

so that $a_\pi \in L^2(|\nu|; \mathcal{H})$. After eventually splitting of singular parts of $\pi$ with respect to $\phi$ as in the proof of Theorem 3.1, the derivative $a_\pi = d\phi/d\pi$ is invertible and Gateaux differentiable with

$$D[a_\pi]\nu = D\left[\left(\frac{d\pi}{d\phi}\right)^{-1}\right]\nu = -\left(\frac{d\pi}{d\phi}\right)^{-2}\frac{d\nu}{d\phi} = -\left(\frac{d\phi}{d\pi}\right)^2\frac{d\nu}{d\phi} = -\frac{d\phi}{d\pi}\frac{d\nu}{d\pi} = -a_\pi \frac{d\nu}{d\pi}.$$

Again using that $d\nu/d\pi$ is bounded, we find $D[a_\pi]\nu \in L^2(|\pi|)$ so that all assumptions of Lemma D.4 are satisfied. The lemma shows that the derivative is of the form $D = I + II + III + IV + V$ and provides formulas for the left hand side. The terms $I$ and $V$ are unchanged in the formula we wish to prove so that it remains to show that $II$, $III$ and $IV$ are zero.

$III$ and $IV$ vanish because both contain the factor $\pi(g_\pi)$, which is zero. Indeed, since $\phi$ is absolutely continuous with respect to $\pi$, the Radon-Nikodym derivative exists and $a_\pi = \frac{d\phi}{d\pi}$. Hence

$$f = \int \sigma(w) \, d\phi(w) = \int \sigma(w)\frac{d\phi}{d\pi}(w) \, d\pi(w) = \int a_\pi(w)\sigma(w) \, d\pi(w) = \pi(a\sigma)$$

and thus in particular $\pi(g_\pi) = 0$.

The remaining term $II$ contains the factor $D_\pi\left[a_\pi(w)\sigma(w)\frac{d\pi}{d\mu}\right]\nu$ for an arbitrary measure $\mu$ with respect to which $\pi$ and $\nu$ are absolutely continuous. The particular choice is not important, e.g. $\mu := |\pi| + |\nu|$ is one possibility. Plugging in $a_\pi = d\phi/d\pi$, we obtain

$$D_\pi\left[a_\pi(w)\sigma(w)\frac{d\pi}{d\mu}\right]\nu = D_\pi\left[\sigma(w)\frac{d\phi}{d\pi}\frac{d\pi}{d\mu}\right]\nu = D_\pi\left[\sigma(w)\frac{d\phi}{d\mu}\right]\nu = 0,$$

where in the last step we have used that the term in the bracket is independent of $\pi$. Hence also $III = 0$, which competes the proof.

$\square$

*Proof of Lemma 3.2.* Let $\nu \in \mathcal{M}_+$ be a positive measure with bounded Radon-Nikodym derivative $d\nu/d\pi$. Then, the local minimizer $\pi$ necessarily satisfies the first order optimality criteria (32)

$$D[\ell(\pi)]\nu + \nu(\lambda) \geq 0,$$

with equality for $\nu = \pi$. Together with the derivative from Lemma E.1, this yields

$$-m \int \|a_\pi(w)\sigma(w)\|^2 \ d\nu(w) + \nu(\lambda) \geq 0,$$

or equivalently

$$\int \left[ -m \|a_\pi(w)\sigma(w)\|^2 + \lambda \right] \frac{d\nu}{d\pi} \ d\pi(w) \geq 0$$

for all non-negative and bounded densities $d\nu/d\pi$, with equality if $d\nu/d\pi = 1$. Thus, it follows that

$$-m \|a_\pi(w)\sigma(w)\|^2 + \lambda = 0,$$

$\pi$-a.e. Dividing by $m$ and redefining $\lambda$, concludes the proof.

$\square$

### E.3    PROOF OF THEOREM 3.3: APPROXIMATION ERROR

By Theorem 3.1, $\pi$ is a global minimum. Let $\pi_B$ be the probability measure from Corollary 2.2. Then, we have

$$\ell(\pi) \leq \ell(\pi_B) \leq \frac{2}{m} |f|_B^2 \,.$$

## F    PROOF OF MAIN RESULTS: AVERAGED OUTER WEIGHTS

The results in this section make heavy use of our conventions on bound and unbound variables summarized in Section A.

### F.1    SYMMETRY

The coefficients $\boldsymbol{a}(\boldsymbol{w})$ in (16) are only confined to a symmetry condition. In this section, we show that this condition is satisfied for least squares minimizers as well as several useful properties for the proofs of the main results.

**Lemma F.1.** *For $f \in \mathcal{H}$, $\sigma\colon \mathcal{W} \to \mathcal{H}$ and $w_i \in \mathcal{W}$, $i \in [M]$, define*

$$\boldsymbol{a} = \boldsymbol{a}(\boldsymbol{w}) = \arg\min_{\boldsymbol{a} \in \mathbb{R}^M}^+ \left\| \frac{1}{M} \sum_{i=1}^M a_i \sigma(\boldsymbol{w}_i) - f \right\|^2,$$

*where $\arg\min^+$ picks the candidate with minimal Euclidean norm, in case the minimizer is not unique. Then for any permutation matrix $P \in \mathbb{R}^{M \times M}$, we have $P\boldsymbol{a}(\boldsymbol{w}) = \boldsymbol{a}(P\boldsymbol{w})$.*

*Proof.* Abbreviating $\sigma_i = \sigma(w_i)$ and solving the least squares problem, $\boldsymbol{a}$ satisfies the linear system of equations

$$A_{\boldsymbol{w}}\boldsymbol{a}(\boldsymbol{w}) = b_{\boldsymbol{w}}, \qquad A_{\boldsymbol{w}} := \left[ \frac{1}{M} \langle \sigma_i, \sigma_j \rangle \right]_{i,j=1}^M, \qquad b_{\boldsymbol{w}} := [\langle \sigma_i, f \rangle]_{i=1}^M \,.$$

Its solution is invariant under permutations. Indeed, for permutation matrix $P \in \mathbb{R}^{M \times M}$, replacing $\boldsymbol{w}$ by $P\boldsymbol{w}$, we have $A_{P\boldsymbol{w}}\boldsymbol{a}(P\boldsymbol{w}) = b_{P\boldsymbol{w}}$. But since $P^T = P^{-1}$ we also have $(PA_{\boldsymbol{w}}P^T)P\boldsymbol{a}(\boldsymbol{w}) = Pb_{\boldsymbol{w}}$, or equivalently $A_{P\boldsymbol{w}}P\boldsymbol{a}(\boldsymbol{w}) = b_{P\boldsymbol{w}}$. Hence

$$P\boldsymbol{a}(\boldsymbol{w}) = \boldsymbol{a}(P\boldsymbol{w}) = A_{P\boldsymbol{w}}^+ b_{P\boldsymbol{w}},$$

which concludes the proof.

$\square$

**Lemma F.2.** *Let $p\colon [M] \to [M]$ be a permutation that leaves the complement $[M] \setminus \Lambda$ of a set $\Lambda \subset [M]$ invariant. Assume that the corresponding permutation matrix satisfies $P\boldsymbol{a}(\boldsymbol{w}) = \boldsymbol{a}(P\boldsymbol{w})$ for some function $\boldsymbol{a}\colon \mathcal{W}^M \to \mathbb{R}$. Let $h\colon \mathcal{W} \to \mathcal{H}$ be a function and $\pi_k \in \mathcal{M}_{+,1}$, $k \in \Lambda$ be probability measures. Then we have*

$$\mathbb{E}_{w_{p_k} \sim \pi_k; k \in \Lambda} \left[ \boldsymbol{a}_{p_i}(\boldsymbol{w}) h(w_{p_i}) \right] = \mathbb{E}_{w_k \sim \pi_k; k \in \Lambda} \left[ \boldsymbol{a}_i(\boldsymbol{w}) h(w_i) \right] \quad \text{for all } i \in \Lambda.$$

*Proof.* Let $E$ be the expectation on the left hand side in the lemma. By the given properties of the permutation, we have

$$\boldsymbol{a}_{p_i}(w_1, \ldots, w_M) = \boldsymbol{a}_i(w_{p_1}, \ldots, w_{p_M}),$$

so that

$$E = \int \boldsymbol{a}_{p_i}(w_1, \ldots, w_M) h(w_{p_i}) \prod_{k \in \Lambda} d\pi_k(w_{p_k})$$

$$= \int \boldsymbol{a}_i(w_{p_1}, \ldots, w_{p_M}) h(w_{p_i}) \prod_{k \in \Lambda} d\pi_k(w_{p_k}).$$

Since $p$ leaves the complement of $\Lambda$ invariant, we can change the integration variables names $w_{p_i} \to w_i$ to obtain

$$E = \int \boldsymbol{a}_i(w_1, \ldots, w_M) h(w_i) \prod_{k \in \Lambda} d\pi_k(w_k),$$

which concludes the proof.

$\square$

**Lemma F.3.** *Let $\Lambda \subset [M]$. Let $w_i \in \mathcal{W}$ for $i \notin \Lambda$ and sample $w_i \in \mathcal{W}$, $i \in \Lambda$ i.i.d. from probability measure $\pi$. Assume $\boldsymbol{a}\colon \mathcal{W}^M \to \mathbb{R}$ satisfies $P\boldsymbol{a}(\boldsymbol{w}) = \boldsymbol{a}(P\boldsymbol{w})$ for all permutation matrices $P \in \mathbb{R}^{M \times M}$ and define $f_{\boldsymbol{a}(\boldsymbol{w}),\boldsymbol{w}} = \frac{1}{M} \sum_{i=1}^{M} \boldsymbol{a}_i(\boldsymbol{w}) \sigma(w_i)$. Then for $\mathbb{E}_\Lambda = \mathbb{E}_{w_i \sim \pi, i \in \Lambda}$, we have*

$$\mathbb{E}_\Lambda \left[ \boldsymbol{a}_i(\boldsymbol{w}) \sigma(w_i) \right] = \mathbb{E}_\Lambda \left[ \boldsymbol{a}_j(\boldsymbol{w}) \sigma(w_j) \right] \quad \text{for all } i, j \in \Lambda \tag{33}$$

*and*

$$\mathbb{E}_\Lambda \left[ \boldsymbol{a}_i(\boldsymbol{w}) \sigma(w_i) \right] = \frac{M}{|\Lambda|} \mathbb{E}_\Lambda \left[ f_{\boldsymbol{a}(\boldsymbol{w}),\boldsymbol{w}} \right] - \frac{1}{|\Lambda|} \sum_{j \notin \Lambda} \mathbb{E}_\Lambda \left[ \boldsymbol{a}_j(\boldsymbol{w}) \sigma(w_j) \right] \quad \text{for all } i \in \Lambda. \tag{34}$$

*Proof.* We abbreviate $\boldsymbol{a}_i = \boldsymbol{a}_i(\boldsymbol{w})$ and $\sigma_i = \sigma(w_i)$. The first identity (33) follows directly from Lemma F.2 with $\pi_k = \pi$, permutation $p$ that swaps $i$ and $j$ and $h = \sigma$. To show (34), we split the indices across $\Lambda$:

$$\mathbb{E}_\Lambda \left[ f_{\boldsymbol{a}(\boldsymbol{w}),\boldsymbol{w}} \right] = \frac{1}{M} \sum_{i \in \Lambda} \mathbb{E}_\Lambda \left[ \boldsymbol{a}_i \sigma_i \right] + \frac{1}{M} \sum_{i \notin \Lambda} \mathbb{E}_\Lambda \left[ \boldsymbol{a}_i \sigma_i \right].$$

By (33) all summands in the sum over $i \in \Lambda$ are equal so that we can choose one index $i \in \Lambda$ and collapse the sum:

$$\mathbb{E}_\Lambda \left[ f_{\boldsymbol{a}(\boldsymbol{w}),\boldsymbol{w}} \right] = \frac{|\Lambda|}{M} \mathbb{E}_\Lambda \left[ \boldsymbol{a}_i \sigma_i \right] + \frac{1}{M} \sum_{i \notin \Lambda} \mathbb{E}_\Lambda \left[ \boldsymbol{a}_i \sigma_i \right], \qquad \text{for all } i \in \Lambda.$$

Rearranging terms shows (34) and concludes the proof.

$\square$

**Corollary F.4.** *Assume $w_i \in \mathcal{W}$, $i \in [M]$ are sampled i.i.d. from a probability measure $\pi$. Assume $\boldsymbol{a}\colon \mathcal{W}^M \to \mathbb{R}$ satisfies $P\boldsymbol{a}(\boldsymbol{w}) = \boldsymbol{a}(P\boldsymbol{w})$ for all permutation matrices $P \in \mathbb{R}^{M \times M}$ and define $f_{\boldsymbol{a}(\boldsymbol{w}),\boldsymbol{w}} = \frac{1}{M} \sum_{i=1}^{M} \boldsymbol{a}_i(\boldsymbol{w}) \sigma(w_i)$. Then we have*

$$\mathbb{E} \left[ \boldsymbol{a}_i(\boldsymbol{w}) \sigma(w_i) \right] = \mathbb{E} \left[ \boldsymbol{a}_j(\boldsymbol{w}) \sigma(w_j) \right] = \mathbb{E} \left[ f_{\boldsymbol{a}(\boldsymbol{w}),\boldsymbol{w}} \right] \qquad \text{for all } i, j \in \Lambda.$$

*Proof.* The result follows directly from Lemma F.3 with $\Lambda = [M]$. Then $M/|\Lambda| = 1$ and the sum over $j \notin \Lambda = [M]$ in (34) vanishes. $\square$

This section provides several derivatives for the main results. We abbreviate

$$g_\pi(w) := \bar{a}_\pi(w)\sigma(w) - f, \tag{35}$$

with $\bar{a}_\pi$ defined in (17).

**Lemma F.5.** *Let $\pi \in \mathcal{M}_{+,1}$, $g_\pi$ be given by (35) and $f_{\boldsymbol{w}}$ be given by (17). Then*

$$\pi(g_\pi) = \mathbb{E}_{\boldsymbol{w} \sim \pi^M}\left[f_{\boldsymbol{w}} - f\right].$$

*Proof.* Since all $w_i$, $i \in [M]$ are distributed by $\pi$, we have

$$
\begin{aligned}
\pi(g_\pi) &= \mathbb{E}_{w_1}\left[\bar{a}(w_1)\sigma(w_1) - f\right] \\
&= \mathbb{E}_{w_1}\left[\mathbb{E}_{\boldsymbol{w}_{\setminus 1}}\left[\boldsymbol{a}_1(\boldsymbol{w})\right]\sigma(w_1) - f\right] \\
&= \mathbb{E}_{w_1}\mathbb{E}_{\boldsymbol{w}_{\setminus 1}}\left[\boldsymbol{a}_1(\boldsymbol{w})\sigma(w_1) - f\right] \\
&= \mathbb{E}_{\boldsymbol{w}}\left[\boldsymbol{a}_1(\boldsymbol{w})\sigma(w_1) - f\right] \\
&= \mathbb{E}_{\boldsymbol{w}}\left[f_{\boldsymbol{a}(\boldsymbol{w}),\boldsymbol{w}} - f\right] \\
&= \mathbb{E}_{\boldsymbol{w}}\left[f_{\boldsymbol{w}} - f\right],
\end{aligned}
$$

where in the first equality we use the definition (35) of $g_\pi$, in the second the definition (17) of $\bar{a}(w_1)$, in the fourth we join expectations, in the fifth we use Corollary F.4 and in the last the definition (17) of $f_{\boldsymbol{w}}$.

$\square$

**Lemma F.6.** *Let $\pi \in \mathcal{M}_{+,1}$, $\boldsymbol{a}_1 \in L^2(\pi^M)$ be defined by (16) and $\bar{a}_\pi$ be defined by (17). Then for any $\nu \ll \pi$ with bounded $d\nu/d\pi$*

$$D_\pi[\bar{a}_\pi(w_1)]\nu = (M-1)\int \mathbb{E}_{\boldsymbol{w}_{\setminus 1, j}}\left[\boldsymbol{a}_1(\boldsymbol{w})\right]\,d\nu(w_j), \qquad \text{for all } j \neq 1.$$

*Proof.* With definition (17) of $\bar{a}_\pi$ applied to input $w_1$, we compute the derivative

$$
\begin{aligned}
D_\pi[\bar{a}_\pi(w_1)]\nu &= D_\pi\left[\int \boldsymbol{a}_1(\boldsymbol{w})\prod_{i=2}^{M} d\pi(w_i)\right] \\
&= \sum_{j=2}^{M}\int \boldsymbol{a}_1(\boldsymbol{w})\prod_{\substack{i=2 \\ i\neq j}}^{M} d\pi(w_i)\,d\nu(w_j) \\
&= \sum_{j=2}^{M}\int \mathbb{E}_{\boldsymbol{w}_{\setminus 1, j}}\left[\boldsymbol{a}_1(\boldsymbol{w})\right]\,d\nu(w_j).
\end{aligned}
$$

Next, we show that the summands are independent of $j$. Indeed, by Lemma F.2, with $\Lambda = 2, \ldots, M$, permutation $p$ that swaps indices $j$ and arbitrary $k \neq$, $h = 1$ and $\pi_k = \nu$, we have

$$\int \mathbb{E}_{\boldsymbol{w}_{\setminus 1, j}}\left[\boldsymbol{a}_1(\boldsymbol{w})\right]\,d\nu(w_j) = \int \mathbb{E}_{\boldsymbol{w}_{\setminus 1, k}}\left[\boldsymbol{a}_1(\boldsymbol{w})\right]\,d\nu(w_k).$$

Therefore, in the sum above, we can choose an arbitrary index $j \neq 1$ and collapse the sum to the factor $M - 1$, which concludes the proof.

$\square$

**Lemma F.7.** *Let $\pi \in \mathcal{M}_{+,1}$, $\boldsymbol{a}_1 \in L^2(\pi^M)$ be defined by (16), $\bar{a}_\pi$ by (17) and $g_\pi = \bar{a}_\pi\sigma - f$. Then for any $\nu \ll \pi$ with bounded $d\nu/d\pi$ and $i \in [M]$*

$$\pi(D_\pi[g_\pi]\nu) = M\nu\left(\mathbb{E}_{\boldsymbol{w}_{\setminus i}}\left[f_{\boldsymbol{w}}\right]\right) - \nu(g_\pi) - \nu(1)f.$$

*Proof.* Using the definition of $g_\pi$ and Lemma F.6, for an arbitrary index $j \neq 1$, we have

$$\pi(D_\pi[g_\pi]\nu) = \mathbb{E}_{w_1} \left[ D_\pi[\bar{a}_\pi(w_1)]\nu\sigma(w_1) \right]$$

$$= (M-1) \int \mathbb{E}_{w_1}\mathbb{E}_{\boldsymbol{w}_{\backslash 1,j}} \left[ \boldsymbol{a}_1(\boldsymbol{w})\sigma(w_1) \right] \, d\nu(w_j)$$

$$= (M-1) \int \mathbb{E}_{\boldsymbol{w}_{\backslash j}} \left[ \boldsymbol{a}_1(\boldsymbol{w})\sigma(w_1) \right] \, d\nu(w_j).$$

Invoking Lemma F.3 with $\Lambda = [M] \setminus \{j\}$, this implies

$$\pi(D_\pi[g_\pi]\nu) = M \int \mathbb{E}_{\boldsymbol{w}_{\backslash j}} \left[ f_{\boldsymbol{a}(\boldsymbol{w}),\boldsymbol{w}} \right] \, d\nu(w_j) - \int \mathbb{E}_{\boldsymbol{w}_{\backslash j}} \left[ \boldsymbol{a}_j(\boldsymbol{w})\sigma(w_j) \right] \, d\nu(w_j)$$

$$= I - II.$$

Using the abbreviation $f_{\boldsymbol{w}} = f_{\boldsymbol{a}(\boldsymbol{w}),\boldsymbol{w}}$, we simplify the first term to

$$I = M \int \mathbb{E}_{\boldsymbol{w}_{\backslash j}} \left[ f_{\boldsymbol{w}} \right] \, d\nu(w_j).$$

Since the $f_{\boldsymbol{w}}$ is invariant under permutations of $\boldsymbol{w}$, we can replace $j$ by an arbitrary index $i \in [M]$, including 1, which was not allowed in the definition of $j$:

$$I = M \int \mathbb{E}_{\boldsymbol{w}_{\backslash i}} \left[ f_{\boldsymbol{w}} \right] \, d\nu(w_i) = M\nu \left( \mathbb{E}_{\boldsymbol{w}_{\backslash i}} \left[ f_{\boldsymbol{w}} \right] \right).$$

To simplify $II$, we use symmetry from Lemma F.2, to rename $1 \leftrightarrow j$ so that

$$II = \int \mathbb{E}_{\boldsymbol{w}_{\backslash 1}} \left[ \boldsymbol{a}_1(\boldsymbol{w})\sigma(w_1) \right] \, d\nu(w_1)$$

$$= \int \mathbb{E}_{\boldsymbol{w}_{\backslash 1}} \left[ \boldsymbol{a}_1(\boldsymbol{w}) \right] \sigma(w_1) \, d\nu(w_1) = \int \bar{a}_\pi(w_1)\sigma(w_1) \, d\nu(w_1),$$

where the last step is the definition (17) of $\bar{a}_\pi$. Inserting $f$, we can express this in terms of $g_\pi$:

$$II = \int \bar{a}_\pi(w_1)\sigma(w_1) - f \, d\nu(w_1) + \int f \, d\nu(w_1),$$

$$= \int g_\pi(w_1) \, d\nu(w_1) + \int f \, d\nu(w_1),$$

$$= \nu(g_\pi) + \nu(1)f.$$

Combining all terms

$$\pi(D_\pi[g_\pi]\nu) = M\nu \left( \mathbb{E}_{\boldsymbol{w}_{\backslash i}} \left[ f_{\boldsymbol{w}} \right] \right) - \nu(g_\pi) - \nu(1)f$$

concludes the proof.

$\square$

The following corollary is a minor rearrangement of Lemma F.7, in the exact form used below.

**Corollary F.8.** *Let $\pi \in \mathcal{M}_{+,1}$, $\boldsymbol{a}_1 \in L^2(\pi^M)$ be defined by (16), $\bar{a}_\pi$ by (17) and $g_\pi = \bar{a}_\pi \sigma - f$. Then for any $\nu \ll \pi$ with bounded $d\nu/d\pi$ and $i \in [M]$*

$$\pi(D_\pi[g_\pi]\nu) = M\nu \left( \mathbb{E}_{\boldsymbol{w}_{\backslash i}} \left[ f_{\boldsymbol{w}} - f_{\boldsymbol{w}_{\backslash i}} \right] \right) + M\nu \left( \mathbb{E}_{\boldsymbol{w}_{\backslash i}} \left[ f_{\boldsymbol{w}_{\backslash i}} \right] \right) - \nu(g_\pi) - \nu(1)f.$$

*Proof.* Follows directly from Lemma F.7 with

$$\mathbb{E}_{\boldsymbol{w}_{\backslash i}} \left[ f_{\boldsymbol{w}} \right] = \mathbb{E}_{\boldsymbol{w}_{\backslash i}} \left[ f_{\boldsymbol{w}} - f_{\boldsymbol{w}_{\backslash i}} \right] + \mathbb{E}_{\boldsymbol{w}_{\backslash i}} \left[ f_{\boldsymbol{w}_{\backslash i}} \right]$$

$\square$

**Lemma F.9.** *Let $\pi \in \mathcal{M}_{+,1}$, $\boldsymbol{a}_1 \in L^2(\pi^M)$ be defined by (16), $\bar{a}_\pi$ by (17), $\Delta_1$ by (19) and $g_\pi = \bar{a}_\pi \sigma - f$. Then for any $\nu \ll \pi$ with bounded $d\nu/d\pi$*

$$\int D_\pi \left[ \|g_\pi(w)\|^2 \right] \nu \, d\pi(w) = \nu(\Delta_1) - 2\nu(\|g_\pi\|^2) - 2 \langle \nu(g_\pi), f \rangle .$$

We motivate the lemma with a simplified analogy, where:

- $\sigma(w) = 1$ and $f = 0$,
- the measures $d\pi(w) = \rho(w)\,dw$ and $d\nu(w) = v(w)\,dw$ and $d\phi(w) = F(w)\,dw$ are given by densities
- and the coefficients by $a_\rho = d\phi/d\pi = F/\rho$ are defined by Radon-Nikodym derivatives as for the exact outer weights (14).

Then the derivative $D_\rho[a_\rho \rho] = D_\rho[F] = 0$ vanishes because the inner function is independent of $\rho$. Thus by the product rule, we have

$$
\begin{aligned}
0 &= 2 \int \langle a_\rho(w), D_\rho[a_\rho(w)\rho(w)]v \rangle\, \rho(w)\,dw \\
&= 2 \int \langle a_\rho(w), D_\rho[a_\rho(w)]v \rangle\, \rho(w)\,dw + 2 \int \langle a_\rho(w), a_\rho(w) \rangle\, D_\rho[\rho(w)]v\,dw \\
&= \int D_\rho \left[ \|a_\rho(w)\|^2 v \right] \rho(w)\,dw + 2 \int \|a_\rho(w)\|^2 v(w)\,dw. \\
&=: d - ii.
\end{aligned}
$$

We can now compare the term in this informal formula with terms in Lemma F.9, split into $D = I + II + III$. Then, $d$ is analogous to $D$ and $ii$ to $II$. The remaining term $I$ is a perturbation and $III$ is a constant, which will be joined with Lagrange multipliers for normalization $\pi(1) = 1$.

The above motivation also allows a direct comparison with the main results for exact outer weights $a_\pi$ in (14). Then, $D_\rho[a_\rho \rho]$ corresponds to the derivative in term II in Lemma D.4. Here, we have used that $D[a_\rho \rho]$ is zero and likewise Lemma 3.2 shows that $II$ is zero, which is a crucial step in the proof of equidistribution Lemma 3.2.

*Proof.* Let $D$ denote the left hand side in the lemma's conclusion. Computing the derivative, plugging in the definition of $g_\pi$ and renaming $w \to w_1$ for convenience below, we obtain

$$
D = 2 \int \langle g_\pi(w), D_\pi[g_\pi(w)]\nu \rangle\, d\pi(w) = 2 \int \langle g_\pi(w_1), D_\pi[\bar{a}_\pi(w_1)]\sigma(w_1)\nu \rangle\, d\pi(w_1).
$$

We compute the derivative of $\bar{a}_\pi(w)$ by Lemma F.6, with an arbitrary $j \neq 1$, and rearrange terms to conclude that

$$
D = 2(M-1) \iint \langle g_\pi(w_1), \mathbb{E}_{\boldsymbol{w}\setminus 1,j} [\boldsymbol{a}_1(\boldsymbol{w})\sigma(w_1)] \rangle\, d\nu(w_j)\, d\pi(w_1).
$$

Next, we swap the variables $w_1$ and $w_j$ and apply symmetry, Lemma F.2 with $h(w) = \langle g_\pi(w), \sigma(w) \rangle$, to obtain

$$
D = 2(M-1) \iint \langle g_\pi(w_j), \mathbb{E}_{\boldsymbol{w}\setminus 1,j} [\boldsymbol{a}_j(\boldsymbol{w})\sigma(w_j)] \rangle\, d\nu(w_1)\, d\pi(w_j).
$$

Since $D$ is independent of $j \neq 1$, so is the right hand side and we can average over all choices:

$$
D = 2 \sum_{j=2}^{M} \iint \langle g_\pi(w_j), \mathbb{E}_{\boldsymbol{w}\setminus 1,j} [\boldsymbol{a}_j(\boldsymbol{w})] \sigma(w_j) \rangle\, d\nu(w_1)\, d\pi(w_j).
$$

Next, we add and subtract the missing $j = 1$ term in the sum:

$$
\begin{aligned}
D = {}& 2 \sum_{j=1}^{M} \iint \langle g_\pi(w_j), \mathbb{E}_{\boldsymbol{w}\setminus 1,j} [\boldsymbol{a}_j(\boldsymbol{w})] \sigma(w_j) \rangle\, d\nu(w_1)\, d\pi(w_j). \\
& - 2 \int \langle g_\pi(w_1), \mathbb{E}_{\boldsymbol{w}\setminus 1} [\boldsymbol{a}_1(\boldsymbol{w})\sigma(w_1)] \rangle\, d\nu(w_1) \\
= {}& I - II.
\end{aligned}
$$

Note that $II$ does not contain a $\pi$ integral because $w_1$ is already integrated with respect to $\nu$ and $\pi(1) = 1$. In order to simplify $II$, we first replace the expectation with the definition of $\bar{a}_\pi$

$$II = 2 \int \langle g_\pi(w_1), \bar{a}_\pi(\boldsymbol{w}_1)\sigma(w_1) \rangle \, d\nu(w_1).$$

We add and subtract $f$ so that we can replace the second component of the inner product with $g_\pi$:

$$II = 2 \int \langle g_\pi(w_1), \bar{a}_\pi(w_1)\sigma(w_1) - f \rangle \, d\nu(w_1) + 2 \int \langle g_\pi(w_1), f \rangle \, d\nu(w_1)$$

$$= 2 \int \|g_\pi(w_1)\|^2 \, d\nu(w_1) + 2 \int \langle g_\pi(w_1), f \rangle \, d\nu(w_1)$$

$$= 2\nu(\|g_\pi\|^2) + 2 \langle \nu(g_\pi), f \rangle.$$

Collecting all terms $D = I - II$ and noting that $I = \nu(\Delta_1)$, given by (19), completes the proof.

$\square$

**Lemma F.10.** *Let* $\pi \in \mathcal{M}_{+,1}$, $\boldsymbol{a}_1 \in L^2(\pi^M)$ *be defined by* (16), $\bar{a}_\pi$ *by* (17), $\Delta_1$ *by* (19) *and* $G_\pi \in \mathcal{H}$ *by* (22). *Then for any* $\nu \ll \pi$ *with bounded* $d\nu/d\pi$

$$\nu(\Delta_1) = 2M \langle G_\pi, \nu(\mathbb{E}_{\boldsymbol{w}_{\setminus 1}}[f_{\boldsymbol{w}}]) \rangle.$$

*Proof.* Recall that $\nu(\Delta_1)$ is defined by

$$I = 2 \sum_{j=1}^{M} \iint \langle g_\pi(w_j), \mathbb{E}_{\boldsymbol{w}_{\setminus 1,j}}[\boldsymbol{a}_j(\boldsymbol{w})] \sigma(w_j) \rangle \, d\nu(w_1) \, d\pi(w_j).$$

Since $\mathbb{E}_{\boldsymbol{w}_{\setminus 1,j}}[\boldsymbol{a}_j(\boldsymbol{w})]$ is scalar, we can factor it out of the inner product to obtain

$$I = 2 \sum_{j=1}^{M} \iint \langle g_\pi(w_j), \sigma(w_j) \rangle \, \mathbb{E}_{\boldsymbol{w}_{\setminus 1,j}}[\boldsymbol{a}_j(\boldsymbol{w})] \, d\nu(w_1) \, d\pi(w_j).$$

and invoke the given assumption of $G_\pi$:

$$I = 2 \sum_{j=1}^{M} \iint \langle G_\pi, \sigma(w_j) \rangle \, \mathbb{E}_{\boldsymbol{w}_{\setminus 1,j}}[\boldsymbol{a}_j(\boldsymbol{w})] \, d\nu(w_1) \, d\pi(w_j).$$

For $j = 1$, the variable $w_j = w_1$ is bound to the $\nu$ integral and the outer $\pi$ integral can be removed. For $j \neq 1$, note that unlike $g_\pi(w_j)$, the new quantity $G_\pi$ is independent of $\boldsymbol{w}$ so that we can move the outer $\pi$ integral inside to join it with the expectation:

$$I = 2 \sum_{j=1}^{M} \int \langle G_\pi, \sigma(w_j) \rangle \, \mathbb{E}_{\boldsymbol{w}_{\setminus 1}}[\boldsymbol{a}_j(\boldsymbol{w})] \, d\nu(w_1).$$

Next, we move the sum inside and abbreviate it by $f_{\boldsymbol{w}}$

$$I = 2M \int \left\langle G_\pi, \mathbb{E}_{\boldsymbol{w}_{\setminus 1}} \left[ \frac{1}{M} \sum_{j=1}^{M} \boldsymbol{a}_j(\boldsymbol{w})\sigma(w_j) \right] \right\rangle \, d\nu(w_1)$$

$$= 2M \int \langle G_\pi, \mathbb{E}_{\boldsymbol{w}_{\setminus 1}}[f_{\boldsymbol{w}}] \rangle \, d\nu(w_1)$$

$$= 2M \langle G_\pi, \nu(\mathbb{E}_{\boldsymbol{w}_{\setminus 1}}[f_{\boldsymbol{w}}]) \rangle,$$

which proves the lemma.

$\square$

The following is a slight rearrangement of Lemma F.9 in the exact format that will be used below.

**Corollary F.11.** *Let* $\pi \in \mathcal{M}_{+,1}$, $\boldsymbol{a}_1 \in L^2(\pi^M)$ *be defined by* (16), $\bar{a}_\pi$ *by* (17) *and* $g_\pi = \bar{a}_\pi \sigma - f$. *Then for any* $\nu \ll \pi$ *with bounded* $d\nu/d\pi$

$$\int D_\pi \left[ \|g_\pi(w)\|^2 \right] \nu \, d\pi(w) = \nu(\Delta_1) + \nu(\bar{\lambda}) - 2\nu(\|g_\pi\|^2) - 2 \left\langle \nu(g_\pi), f \right\rangle.$$

*with*

$$\bar{\lambda} = 0, \qquad\qquad\qquad \text{if } \Delta_1 \text{ is defined by (19)},$$
$$\bar{\lambda} = 2M \left\langle G_\pi, \mathbb{E}_{\boldsymbol{w}_{\backslash 1}} \left[ f_{\boldsymbol{w}_{\backslash 1}} \right] \right\rangle, \qquad\qquad \text{if } \Delta_1 \text{ is defined by (20)}.$$

*Proof.* In case $\Delta_1$ is defined by (19), there is nothing to show. If it is defined by (20), by Lemma F.10, we have

$$\nu(\Delta_1) = 2M \left\langle G_\pi, \nu(\mathbb{E}_{\boldsymbol{w}_{\backslash 1}} [f_{\boldsymbol{w}}]) \right\rangle$$
$$= 2M \left\langle G_\pi, \nu(\mathbb{E}_{\boldsymbol{w}_{\backslash 1}} \left[ f_{\boldsymbol{w}} - f_{\boldsymbol{w}_{\backslash 1}} \right]) \right\rangle + 2M \left\langle G_\pi, \nu(\mathbb{E}_{\boldsymbol{w}_{\backslash 1}} \left[ f_{\boldsymbol{w}_{\backslash 1}} \right]) \right\rangle.$$

On the right hand side, the first term matches the alternative definition (20) of $\Delta_1$. The second simplifies to

$$2M \left\langle G_\pi, \mathbb{E}_{\boldsymbol{w}_{\backslash 1}} \left[ f_{\boldsymbol{w}_{\backslash 1}} \right] \right\rangle \nu(1) = \bar{\lambda}\nu(1),$$

because the inner expectation $\mathbb{E}_{\boldsymbol{w}_{\backslash 1}} \left[ f_{\boldsymbol{w}_{\backslash 1}} \right]$ does not depend on the integration variable $w_1$ of the $\nu$ integral. This concludes the proof.

$\square$

**Lemma F.12.** *Let* $\pi \in \mathcal{M}_{+,1}$, $\boldsymbol{a}_1 \in L^2(\pi^M)$ *be defined by* (16), $\bar{a}_\pi$ *by* (17), $\Delta_1$ *by* (19) *or* (20) *and* $\Delta_2$ *by* (21). *Then there is a constant* $\lambda \in \mathbb{R}$ *so that for all* $\nu \ll \pi$ *with bounded* $d\nu/d\pi$ *the derivative of the loss* (18) *is*

$$D_\pi \left[ \ell(\pi) \right] \nu = -m\nu(\|\bar{a}_\pi \sigma\|^2) + m\lambda\nu(1) + m\nu(\Delta_1) + m\nu(\Delta_2).$$

*Proof.* With $g_\pi = \bar{a}_\pi \sigma - f$ and $\boldsymbol{g}_\pi = \sum_{i=1}^m g_\pi(w_i)$ by the product rule, we have

$$D_\pi \left[ \ell(\pi) \right] \nu = D_\pi \left[ \int \|\boldsymbol{g}_\pi\|^2 \, d\pi^n(\boldsymbol{w}) \right] \nu$$
$$= D_\pi \left[ \int \|\boldsymbol{g}_\eta\|^2 \, d\pi^n(\boldsymbol{w}) \right] \nu \Big|_{\eta=\pi} + \int D_\pi \left[ \|\boldsymbol{g}_\pi\|^2 \right] \nu \, d\pi^n(\boldsymbol{w}).$$

The first term, for fixed inner $\eta = \pi$, not differentiated, is given by Lemma D.2 and the second by Lemma D.3. With normalization $\pi(1) = 1$, this yields

$$D_\pi \left[ \ell(\pi) \right] \nu = I + II + III + IV + V$$

with

$$I = m\nu(\|g_\pi\|^2),$$
$$II = 2m(m-1) \left\langle \pi(g_\pi), \nu(g_\pi) \right\rangle,$$
$$III = \lambda\nu(1),$$
$$IV = m \int D_\pi \left[ \|g_\pi(w)\|^2 \right] \nu \, d\pi(w),$$
$$V = 2m(m-1) \left\langle \pi(g_\pi), \pi(D_\pi[g_\pi]\nu) \right\rangle.$$

By Corollary F.11, we have $IV = IV.1 + IV.2 + IV.3 + IV.4$ with

$$IV.1 = m\nu(\Delta_1),$$
$$IV.2 = m\nu(\bar{\lambda}),$$
$$IV.3 = -2m\nu(\|g_\pi\|^2),$$
$$IV.4 = -2m \left\langle \nu(g_\pi), f \right\rangle.$$

By Lemma F.5 and Corollary F.8, we have $V = V.1 + V.2 + V.3 + V.4$ with

$$V.1 = 2m(m-1)M \left\langle \mathbb{E}_{\boldsymbol{w}} [f_{\boldsymbol{w}} - f], \nu \left( \mathbb{E}_{\boldsymbol{w}_{\setminus 1}} [f_{\boldsymbol{w}} - f_{\boldsymbol{w}_{\setminus 1}}] \right) \right\rangle,$$

$$V.2 = 2m(m-1)M \left\langle \pi(g_\pi), \nu \left( \mathbb{E}_{\boldsymbol{w}_{\setminus 1}} [f_{\boldsymbol{w}_{\setminus 1}}] \right) \right\rangle,$$

$$V.3 = -2m(m-1) \left\langle \pi(g_\pi), \nu(g_\pi) \right\rangle,$$

$$V.4 = -2m(m-1) \left\langle \pi(g_\pi), \nu(1)f \right\rangle.$$

The parts of $III$, $IV.2$, $V.2$, $V.4$ inside of $\nu(\cdot)$ are independent of $w_1$ and therefore can be joined into one constant $\lambda_1$:

$$III + IV.2 + V.2 = \lambda_1 \nu(1).$$

The terms $II$ and $V.3$ cancel:

$$II + V.3 = 0.$$

Next, we join the terms $I$, $IV.3$ and $IV.4$. Using $g_\pi = \bar{a}_\pi \sigma - f$ and defining $\lambda_2 := m\|f\|^2$, we have:

$$\begin{aligned}
I + IV.3 + IV.4 &= -m\nu(\|g_\pi\|^2) - 2m \left\langle \nu(g_\pi), f \right\rangle \\
&= -m\nu(\|\bar{a}_\pi \sigma\|^2) + 2m\nu(\langle \bar{a}_\pi \sigma, f \rangle) - m\nu(\|f\|^2) - 2m \left\langle \nu(\bar{a}_\pi \sigma), f \right\rangle + 2m\nu(\|f\|^2) \\
&= -m\nu(\|\bar{a}_\pi \sigma\|^2) + m \|f\|^2 \nu(1) \\
&= -m\nu(\|\bar{a}_\pi \sigma\|^2) + \lambda_2 \nu(1).
\end{aligned}$$

The only remaining terms are $IV.1 = m\nu(\Delta_1)$ and $V.1 = m\nu(\Delta_2)$, which are contained unchanged in the statement of the lemma. Defining $m\lambda = \lambda_1 + \lambda_2$ and combining all terms concludes the proof.

$\square$

### F.3 STABLE BARRON NORMS AND MAUREY SAMPLING

The following corollary is a variant of standard Maurey sampling arguments in Corollary 2.2, applied to stable Barron norms.

**Corollary F.13.** *Assume probability measure $\pi \in \mathcal{M}_{+,1}$ and $a_\pi \in L^2(\pi)$ satisfy the bounds* (23) (24) *for some $\pi$ integrable function $\delta$ and $\epsilon \in \mathbb{R}$. Then for all $m \in \mathbb{N}$*

$$\mathbb{E}_{\boldsymbol{w} \sim \pi^m} \left\| \frac{1}{m} \sum_{i=1}^m a_\pi(w_i) \sigma(w_i) - f \right\|^2 \le \frac{4}{m} [|f|^2_{B(\delta,\epsilon)} + \pi(\delta)] + 2\epsilon^2.$$

*Proof.* Throughout this proof, let $g_\pi(w) = a_\pi(w)\sigma(w) - f$, for arbitrary $a_\pi \colon \mathcal{W} \to \mathbb{R}$, not only $a_\pi = \bar{a}_\pi$ as in our usual convention. Let $w$, $w_i$ and $w'_i$, $i \in [m]$ be i.i.d. sampled from $\pi$. Since $g_\pi(w) - \pi(g_\pi)$ has zero mean, by Lemma C.1 we have

$$\begin{aligned}
\mathbb{E}_{\boldsymbol{w}} \left\| \frac{1}{m} \sum_{i=1}^m g_\pi(w_i) - \pi(g_\pi) \right\|^2 &= \frac{1}{m^2} \sum_{i=1}^m \mathbb{E}_{\boldsymbol{w},\boldsymbol{w}'} \|g_\pi(w_i) - g_\pi(w'_i)\|^2 \\
&\le \frac{2}{m^2} \sum_{i=1}^m \mathbb{E}_{\boldsymbol{w}} \|a_\pi(w_i)\sigma(w_i)\|^2 = \frac{2}{m} \mathbb{E}_{\boldsymbol{w}} \|a_\pi(w)\sigma(w)\|^2,
\end{aligned}$$

which implies

$$\mathbb{E}_{\boldsymbol{w}} \left\| \frac{1}{m} \sum_{i=1}^m g_\pi(w_i) \right\|^2 \le \frac{4}{m} \mathbb{E}_{\boldsymbol{w}} \|a_\pi(w)\sigma(w)\|^2 + 2 \|\pi(g_\pi)\|^2.$$

From (23) and (24), we have $|a_\pi(w)|^2 \|\sigma(w)\|^2 \le \lambda + \delta(w)$ and $\|\pi(g_\pi)\|^2 \le \epsilon^2$ so that

$$\mathbb{E}_{\boldsymbol{w}} \left\| \frac{1}{m} \sum_{i=1}^m g_\pi(w_i) \right\|^2 \le \frac{4}{m} [\lambda + \pi(\delta)] + 2\epsilon^2.$$

Since $a_\pi$ and $\pi$ are eligible candidates in the definition of the stable Barron norm, we have $\lambda \le |f|_{B(\delta,\epsilon)}$, which concludes the proof.

$\square$

### F.4 PROOF OF LEMMA 3.4: EQUIDISTRIBUTION

We heavily rely on the notational conventions in Appendix A.2. Let $\nu \in \mathcal{M}_+$ be a positive measure with bounded Radon-Nikodym derivative $d\nu/d\pi$. Then, the local minimizer $\pi$ necessarily satisfies the first order optimality criteria (32)

$$D_\pi[\ell(\pi)]\nu + \nu(\lambda) \geq 0,$$

with equality for $\nu = \pi$. With $\Delta_1$ and $\Delta_2$ defined in (19) or (20) and (21), respectively, and $D_\pi[\ell(\pi)]\nu$ computed by Lemma F.12, we obtain upon a redefinition of the constant $\lambda \in \mathbb{R}$

$$-m\nu(\|\bar{a}_\pi\sigma\|^2) + m\lambda\nu(1) + m\nu(\Delta_1) + m\nu(\Delta_2) \geq 0.$$

Equivalently, we have

$$\int \left[ -m\|\bar{a}_\pi(w)\sigma(w)\|^2) + m\lambda + m\Delta_1(w) + m\Delta_2(w) \right] \frac{d\nu}{d\pi}\, d\pi(w) \geq 0,$$

for all non-negative and bounded densities $d\nu/d\pi$, with equality if $d\nu/d\pi = 1$. Thus, it follows that

$$\|\bar{a}_\pi\sigma\|^2 = \lambda + \Delta_1 + \Delta_2$$

$\pi$-almost surely, as stated in the lemma.

### F.5 PROOF OF THEOREM 3.5: APPROXIMATION ERROR

We heavily rely on the notational conventions in Appendix A.2. For arbitrary $\bar{\lambda}$ and $\delta := \bar{\lambda} + \Delta_1 + \Delta_2$, by Lemma 3.4, we have

$$|\bar{a}(w)|^2\|\sigma(w)\|^2 = \lambda + \delta(w) \qquad\qquad \pi - \text{a.s.}$$

for some $\lambda \in \mathbb{R}$ and thus

$$\left| \lambda - |\bar{a}(w)|^2\|\sigma(w)\|^2 \right| \leq |\delta(w)|, \qquad\qquad \pi - \text{a.s.}$$

This establishes the first perturbation bound (23) in the definition of stable Barron norms. By Lemma F.5 the second perturbation bound (24) is satisfied with $\|\mathbb{E}_{w\sim\pi}[g_\pi]\| = \|\pi(g_\pi)\| = \|\mathbb{E}_{\boldsymbol{w}\sim\pi}[f_{\boldsymbol{w}} - f]\| = \epsilon$. Hence, the result follows from Corollary F.13.

## G TRAINING

### G.1 PROOF OF LEMMA 4.1: PARTICLE APPROXIMATION OF WGF

The proof is standard (Chizat and Bach, 2018) and only included to make the exposition self contained. We first compute the gradient in the gradient flow of $\boldsymbol{w}(t)$:

$$\begin{aligned}
\nabla_{w_i}\ell(\pi_{\boldsymbol{w}}) &= \frac{1}{\mathfrak{m}}D[\ell(\pi_{\boldsymbol{w}})]\nabla_{w_i}\delta_{w_i} \\
&= \frac{1}{\mathfrak{m}}\int \nabla_\pi\ell(\pi_{\boldsymbol{w}})(w)\nabla_{w_i}\delta(w - w_i)\, dw \\
&= \frac{1}{\mathfrak{m}}\int \nabla_w\nabla_\pi\ell(\pi_{\boldsymbol{w}})(w)\delta(w - w_i)\, dw \\
&= \frac{1}{\mathfrak{m}}\nabla_w\nabla_\pi\ell(\pi_{\boldsymbol{w}})(w_i),
\end{aligned}$$

where in the third equality we have used integration by parts and the compact support of $\delta$. Hence, the gradient flow of $\boldsymbol{w}(t)$ is equivalent to

$$\dot{w}_i(t) = -\mathfrak{m}\nabla_{w_i}\ell(\pi_{\boldsymbol{w}(t)}) = -\nabla_w\nabla_\pi\ell(\pi_{\boldsymbol{w}(t)})(w_i(t)).$$

Next, we show that $\pi_{\boldsymbol{w}(t)}$ satisfies Wasserstein gradient flow, i.e. that

$$\iint \varphi_t - \nabla_w\varphi(t,w)\nabla_w\nabla_\pi\ell(\pi_{\boldsymbol{w}(t)})(w_i)\, d\pi_{\boldsymbol{w}(t)}\, dt = 0$$

for all $\varphi(t, w)$ with compact support. Using that $\pi_{\boldsymbol{w}(t)}$ consists of Dirac deltas, we can remove the inner integral to obtain

$$\frac{1}{\mathfrak{m}} \sum_{i=1}^{\mathfrak{m}} \int \varphi_t - \nabla_w \varphi(t, w_i(t)) \nabla_w \nabla_\pi \ell(\pi_{\boldsymbol{w}(t)})(w_i(t)) \, dt = 0.$$

Plugging in the identity for $\dot{w}_i(t)$ above, this simplifies to

$$\frac{1}{\mathfrak{m}} \sum_{i=1}^{\mathfrak{m}} \int \varphi_t + \partial_w \varphi(t, w_i(t)) \dot{w}_i(t) \, dt = \frac{1}{\mathfrak{m}} \sum_{i=1}^{\mathfrak{m}} \int \frac{d}{dt} \varphi(t, w_i(t)) \, dt = 0,$$

where the last term is indeed zero because $\varphi$ is compactly supported. This concludes the proof.

## G.2   PROOF OF LEMMA 4.2

We first characterize stationary points of Wasserstein gradient flow.

**Lemma G.1.** *Let $\pi$ be a stationary point of the Wasserstein gradient flow* (25) *with bounded support. Then $\nabla_w \nabla_\pi \ell(\pi) = 0$ $\pi$-almost surely.*

*Proof.* Let $\phi(t) \geq 0$ be compactly supported and normalized $\int \phi(t) \, dt = 1$ and $\psi(w)$ be compactly supported, and equal to one on the support of $\pi$. Then, plugging $\varphi(t, w) = \phi(t)\psi(w)\nabla_\pi \ell(\pi)(w)$ into the distributional definition of WGF and using that $\pi$ is stationary and therefore independent of $t$, we obtain

$$0 = \int \dot{\phi}(t) \, dt \int \psi(w) \nabla_\pi \ell(\pi)(w) \, d\pi(w)$$
$$- \int \phi(t) \, dt \int \nabla_w [\psi(w) \nabla_\pi \ell(\pi)(w)] \cdot \nabla_w \nabla_\pi \ell(\pi)(w) \, d\pi(w) \, dt.$$

The first summand vanishes because $\phi$ has compact support so that $\int \dot{\phi}(t) \, dt = 0$. For the second, we simplify $\psi(w) = 1$ on the support of $\pi$, and with $\int \phi(t) \, dt = 1$, we obtain

$$\int |\nabla_w \nabla_\pi \ell(\pi)(w)|^2 \, d\pi(w) \, dt = 0.$$

It follows that $\nabla_w \nabla_\pi \ell(\pi)(w) = 0$, $\pi$-almost surely.

$\square$

*Proof of Lemma 4.2.* Recall the notational conventions in Appendix A.2. By Lemma F.12, the directional derivatives of the loss are given by

$$D_\pi [\ell(\pi)] \nu = \int -m \|\bar{a}_\pi(w)\sigma(w)\|^2) + m\lambda + m\Delta_1(w) + m\Delta_2(w) \, d\nu(w)$$

so that by our conventions the gradient is

$$\nabla_\pi \ell(\pi)(w) = -\mathfrak{m} \|\bar{a}_\pi(w)\sigma(w)\|^2 + \mathfrak{m}\lambda + \mathfrak{m}\Delta_1(w) + \mathfrak{m}\Delta_2(w),$$

$\pi$-a.e. By Lemma 4.1 the discrete measure $\pi_{\boldsymbol{w}}$ is a stationary point of Wasserstein gradient flow. Then by Lemma G.1, we have $\nabla_w \nabla_\pi \ell(\pi) = 0$ $\pi$-almost surely and thus

$$\nabla_w \left[ -\|\bar{a}_\pi(w)\sigma(w)\|^2 + \Delta_1(w) + \Delta_2(w) \right] = 0,$$

$\pi$-a.e. Since $\pi_{\boldsymbol{w}}$ is a sum of Dirac deltas at location $w_i$, this implies the lemma.

$\square$

## G.3 Proof of (29)

In this section, we compute (29). To this end, we abbreviate

$$\tilde{\ell}(\pi) = \frac{1}{N_\ell} \sum_{j=1}^{N_\ell} L(\boldsymbol{v}^j) := \frac{1}{N_\ell} \sum_{j=1}^{N_\ell} \left\| \frac{1}{m} \sum_{i=1}^{m} \tilde{a}_\pi(v_i^j) \sigma(v_i^j) - f \right\|^2, \qquad \boldsymbol{v}^j \sim \pi^m$$

and recall that gradient descent is given by

$$\boldsymbol{w}^n = \boldsymbol{w}^{n-1} - \gamma \nabla_{\boldsymbol{w}} \tilde{\ell}(\pi_{\boldsymbol{w}^{n-1}}),$$

or plugging in all intermediate steps as well as $\tilde{\ell}(\pi)$ by

$$\boldsymbol{w}^n = \boldsymbol{w}^0 - \gamma \sum_{k=0}^{n-1} \nabla_{\boldsymbol{w}} \tilde{\ell}(\pi_{\boldsymbol{w}^k}) = \boldsymbol{w}^0 - \frac{\gamma}{N_\ell} \sum_{k=0}^{n-1} \sum_{j=1}^{N_\ell} \nabla_{\boldsymbol{w}} L(\boldsymbol{v}^{jk}), \qquad \boldsymbol{v}^{jk} \sim \pi_{\boldsymbol{w}^k}^m.$$

Instead of sampling $\boldsymbol{v}^{jk} \sim \pi_{\boldsymbol{w}^k}^m$ for all $j$, repeatedly from the same distribution, we always sample from the latest one available. To this end, we denote by $l$ the lexicographic ordering the index pairs $jk$ and by $\mathfrak{l}$ the upper bound $k = n$ and $j = N_\ell$ so that

$$\boldsymbol{w}^{\mathfrak{l}} = \boldsymbol{w}^0 - \frac{\gamma}{N_\ell} \sum_{l=0}^{\mathfrak{l}-1} \nabla_{\boldsymbol{w}} L(\boldsymbol{v}^l) \qquad \boldsymbol{v}^l \sim \pi_{\boldsymbol{w}^l}^m.$$

Rewriting as a recursive formula

$$\boldsymbol{w}^{\mathfrak{l}} = \boldsymbol{w}^{\mathfrak{l}-1} - \frac{\gamma}{N_\ell} \nabla_{\boldsymbol{w}} L(\boldsymbol{v}^{\mathfrak{l}-1}) \qquad \boldsymbol{v}^{\mathfrak{l}-1} \sim \pi_{\boldsymbol{w}^{\mathfrak{l}-1}}^m.$$

this yields the gradient descent iteration in (29).

## G.4 Proof of (30)

In this section, we show (30). To this end, we abbreviate

$$L(\boldsymbol{a}, \boldsymbol{w}) := \left\| \frac{1}{M} \sum_{i=1}^{M} a_i^n \sigma(w_i) - f \right\|^2$$

and compute the best approximation coefficients $\boldsymbol{a}(\boldsymbol{w})$ in (15) by gradient descent

$$\boldsymbol{a}^{n+1}(\boldsymbol{w}) = \boldsymbol{a}^n(\boldsymbol{w}) - \lambda \nabla_{\boldsymbol{a}} L(\boldsymbol{a}^n, \boldsymbol{w}).$$

Summing the first component $\boldsymbol{a}_1$ over samples $\boldsymbol{w} = (v, \boldsymbol{v}_{\backslash 1}^j)$ with $\boldsymbol{v}^j \sim \pi^M$, we obtain

$$\frac{1}{N_a} \sum_{j=1}^{N_a} \boldsymbol{a}_1^{n+1}(v, \boldsymbol{v}_{\backslash 1}^j) = \frac{1}{N_a} \sum_{j=1}^{N_a} \boldsymbol{a}_1^n(v, \boldsymbol{v}_{\backslash 1}^j) - \lambda \nabla_{\boldsymbol{a}_1} \frac{1}{N_a} \sum_{j=1}^{N_a} L(\boldsymbol{a}^n, v, \boldsymbol{v}_{\backslash 1}^j)$$

Abbreviating the left hand side by $a_\pi^n(v)$, it converges to the mean

$$\tilde{a}_\pi^n(v) \to \tilde{a}_\pi(v) = \frac{1}{N_a} \sum_{j=1}^{N_a} \boldsymbol{a}_1^\infty(v, \boldsymbol{v}_{\backslash 1}^j),$$

defined in (27). To ease the computation, we replace the argument $\boldsymbol{a}_i^n$ of $L$ with $\tilde{a}_\pi^n(v_i^j)$ to obtain

$$\tilde{a}_\pi^n(v) = \tilde{a}_\pi^{n-1}(v) - \lambda \nabla_a \frac{1}{N_a} \sum_{j=1}^{N_a} L(\tilde{a}_\pi^{n-1}(v, \boldsymbol{v}_{\backslash 1}^j), v, \boldsymbol{v}_{\backslash 1}^j).$$

Note that by our notational conventions the application of $\tilde{a}_\pi^n$ to the vector $(v, \boldsymbol{v}_{\backslash 1}^j)$ is component wise and that $v$ belongs to the discrete set $w_1, \ldots, w_{\mathsf{m}}$ of all particles in the particle approximation (26). Analogous to Appendix G.3, we can always use the newest possible $\tilde{a}_\pi^n$ in the iteration and relabel $n$ to obtain

$$\tilde{a}_\pi^n(v) = \tilde{a}_\pi^{n-1}(v) - \frac{\lambda}{N_a} \nabla_a L(\tilde{a}_\pi^{n-1}(v, \boldsymbol{v}_{\backslash 1}), v, \boldsymbol{v}_{\backslash 1}), \qquad \boldsymbol{v} \sim \pi^M,$$

which provides (30).

# H  PROOFS: PERTURBATIONS

## H.1  PRELIMINARIES

**Lemma H.1.** *Let $\boldsymbol{a}$ and $f_{\boldsymbol{w}}$ be given by (15) and $\mathcal{H}(\boldsymbol{w})$ by (31). Then, for all $\boldsymbol{w} \in \mathcal{W}^M$ and $h \in \mathcal{H}(\boldsymbol{w}_{\setminus 1})$, we have*

$$\left\langle f_{\boldsymbol{w}} - f_{\boldsymbol{w}_{\setminus 1}}, h \right\rangle = 0.$$

*Proof.* Since $f_{\boldsymbol{w}}$ and $f_{\boldsymbol{w}_{\setminus 1}}$ are best approximations, we have

$$\langle f_{\boldsymbol{w}}, h \rangle = \langle f, h \rangle, \qquad \text{for all } h \in \mathcal{H}(\boldsymbol{w}),$$
$$\langle f_{\boldsymbol{w}_{\setminus 1}}, h \rangle = \langle f, h \rangle, \qquad \text{for all } h \in \mathcal{H}(\boldsymbol{w}_{\setminus 1})$$

so that in particular $\left\langle f_{\boldsymbol{w}} - f_{\boldsymbol{w}_{\setminus 1}}, h \right\rangle = 0$ for all $h \in \mathcal{H}(\boldsymbol{w}_{\setminus 1})$.

$\square$

## H.2  PROOF OF LEMMA B.1

From Lemma H.1, we have

$$\left\langle f_{\boldsymbol{w}} - f_{\boldsymbol{w}_{\setminus 1}}, h \right\rangle = 0 \qquad \text{for all } h \in \mathcal{H}(\boldsymbol{w}_{\setminus 1})$$

so that $f_{\boldsymbol{w}_{\setminus 1}}$ is not only a best approximation of $f$ but also of $f_{\boldsymbol{w}}$, i.e.

$$f_{\boldsymbol{w}_{\setminus 1}} = \underset{\varphi \in \mathcal{H}(\boldsymbol{w}_{\setminus 1})}{\arg\min} \left\| \phi - f_{\boldsymbol{w}} \right\|^2.$$

Since $M^{-1}\boldsymbol{a}_1(\boldsymbol{w})$ is defined as a coefficient of the best approximation, we have $f_{\boldsymbol{w}} - M^{-1}\boldsymbol{a}_1(\boldsymbol{w})\sigma(w_1) \in \mathcal{H}(\boldsymbol{w}_{\setminus 1})$ and thus

$$\left\| f_{\boldsymbol{w}} - f_{\boldsymbol{w}_{\setminus 1}} \right\| \leq \left\| f_{\boldsymbol{w}} - [f_{\boldsymbol{w}} - M^{-1}\boldsymbol{a}_1(\boldsymbol{w})\sigma(w_1)] \right\| = M^{-1} \left\| \boldsymbol{a}_1(\boldsymbol{w})\sigma(w_1) \right\|.$$

Applying the expectation on both sides and factoring out terms independent of $\boldsymbol{w}_{\setminus 1}$, proves the lemma.

## H.3  PROOF OF COROLLARY B.2

The result is a direct consequence of Corollary F.13. To this end, we first bound the perturbation terms $\Delta_1$ and $\Delta_2$ given in (20) and (21), respectively. With the given assumptions, we have

$$\mathbb{E}_{\boldsymbol{w}_{\setminus 1}} \left[ |\boldsymbol{a}_1(\boldsymbol{w})| \right] \leq c |\mathbb{E}_{\boldsymbol{w}_{\setminus 1}} \left[ \boldsymbol{a}_1(\boldsymbol{w}) \right]| = |\bar{a}_\pi(w_1)|$$

and therefore, with Lemma B.1

$$\left\| \mathbb{E}_{\boldsymbol{w}_{\setminus 1}} \left[ f_{\boldsymbol{w}} - f_{\boldsymbol{w}_{\setminus 1}} \right] \right\| \leq \mathbb{E}_{\boldsymbol{w}_{\setminus 1}} \left\| f_{\boldsymbol{w}} - f_{\boldsymbol{w}_{\setminus 1}} \right\| \leq M^{-1} \mathbb{E}_{\boldsymbol{w}_{\setminus 1}} \left[ |\boldsymbol{a}_1(\boldsymbol{w})| \right] \left\| \sigma(w_1) \right\|$$
$$\leq c M^{-1} |\bar{a}_\pi(w_1)| \left\| \sigma(w_1) \right\|.$$

Together with

$$\left\langle h, \nu (\mathbb{E}_{\boldsymbol{w}_{\setminus 1}} \left[ f_{\boldsymbol{w}} - f_{\boldsymbol{w}_{\setminus 1}} \right]) \right\rangle = 0$$

for arbitrary $h \in \mathcal{H}(\boldsymbol{w}_{\setminus 1})$ by Lemma H.1, we bound the first perturbation term (20) by

$$\nu(\Delta_1) = 2M \left\langle G_\pi, \nu(\mathbb{E}_{\boldsymbol{w}_{\setminus 1}} \left[ f_{\boldsymbol{w}} - f_{\boldsymbol{w}_{\setminus 1}} \right]) \right\rangle$$
$$= 2M \left\langle G_\pi - h, \nu(\mathbb{E}_{\boldsymbol{w}_{\setminus 1}} \left[ f_{\boldsymbol{w}} - f_{\boldsymbol{w}_{\setminus 1}} \right]) \right\rangle$$
$$\leq 2c \left\| G_\pi - h \right\| \nu(|\bar{a}_\pi(w_1)| \left\| \sigma(w_1) \right\|).$$

Similarly, the second perturbation term (21) is bounded by

$$\nu(\Delta_2) = 2(m-1)M \left\langle \mathbb{E}_{\boldsymbol{w}} \left[ f_{\boldsymbol{w}} - f \right], \nu \left( \mathbb{E}_{\boldsymbol{w}_{\setminus 1}} \left[ f_{\boldsymbol{w}} - f_{\boldsymbol{w}_{\setminus 1}} \right] \right) \right\rangle,$$
$$\leq 2c(m-1) \left\| \mathbb{E}_{\boldsymbol{w}} \left[ f_{\boldsymbol{w}} - f \right] \right\| \nu(|\bar{a}_\pi(w_1)| \left\| \sigma(w_1) \right\|).$$

Since $\nu \ll \pi$ is arbitrary, with $\delta := \Delta_1 + \Delta_2$ we conclude that

$$
\begin{aligned}
|\delta(w)| &\leq 2c \, \|G_\pi - h\| \, |\bar{a}_\pi(w)| \, \|\sigma(w)\| \\
&\quad + 2c(m-1) \, \|\mathbb{E}_{\boldsymbol{w}} [f_{\boldsymbol{w}} - f]\| \, |\bar{a}_\pi(w)| \, \|\sigma(w)\| . \\
&\leq 2c \Big[ \|G_\pi - h\| + (m-1) \, \|\mathbb{E}_{\boldsymbol{w}} [f_{\boldsymbol{w}} - f]\| \Big] \|\bar{a}_\pi \sigma\|_{L^\infty(\pi;\mathcal{H})} \\
&=: \Delta.
\end{aligned}
$$

Next, we establish the bounds (23) and (24) for the definition of stable Barron norms. By Lemma 3.4, we have

$$
|\bar{a}_\pi(w)|^2 \, \|\sigma(w)\|^2 = \lambda + \delta(w) \qquad\qquad \pi - \text{a.s.}
$$

for some $\lambda \in \mathbb{R}$. Combining the last two results yields

$$
\left| \lambda - |\bar{a}_\pi(w)|^2 \, \|\sigma(w)\|^2 \right| \leq \Delta(w),
$$

which shows (23). By Lemma F.5 the second perturbation bound (24) is satisfied with $\|\mathbb{E}_{w\sim\pi} [g_\pi]\| = \|\pi(g_\pi)\| = \|\mathbb{E}_{\boldsymbol{w}\sim\pi} [f_{\boldsymbol{w}} - f]\| = \epsilon$. Hence, the result follows from Corollary F.13.

