# OpenReview forum: "Barron Approximation and locally optimal weight densities for shallow neural networks"
_ICLR.cc/2026/Conference — Submitted to ICLR 2026_

### Official Review · Reviewer_xqKn · 2025-10-27

**Soundness:** 3
**Presentation:** 1
**Contribution:** 2
**Rating:** 4
**Confidence:** 3

**Summary:**

This paper views the finite-width one-hidden-layer MLP as an approximation of its infinite limit with the integral form. Setting up the target function as an integral, the paper studies the approximation error using a finite-sized network. Based on the specific parameterization, the paper shows several main results: $i)$. the optimization landscape of the probability measure is convex; $ii)$. the local minimizers of the objective satisfies equidistribution property, i.e. the norm of the integrand are independent of the sample; and $iii)$. second-layer weights can be explicitly constructed from the integral form. Lastly, the paper also studied Wasserstein gradient flow and regular gradient descent within the particle approximation, and illustrated the connection with dropout.

**Strengths:**

1. The paper provided a detailed study of approximating the target function using the finite-sized network, and derived the corresponding approximation based on a newly defined stable Barron norm.
2. The paper demonstrated a way to construct the second layer weight given a ideal Radon-Nikodym derivative.
3. The paper derive the form of gradient descent by approximating the expectations in the objective with a sample mean, and shows the connection between gradient descent and dropout.

**Weaknesses:**

1. By its current form, the paper is hard to be appreciated by a general audience. In particular, the paper starts from a particular parameterization of the neural network in Eq. (1) and does not motivate why this particular parameterization is chosen. In particular, it is unclear why the second layer weight $a_{\pi}(w)$ need to depend on the first layer weight $w$. Moreover, the paper set up the target function in Eq. (7) and argues that the measure $\phi(w)$ is hard to estimate because it is a signed measure, but it is unclear whether the target function defined over a signed measure is something that have practical interest. Lastly, the significance of the derived results, including the approximation bound depending on the stable Barron measure, the equidistribution property, and the convexity of the objective, are vague. In general, the paper lacks a motivation of its mathematical setup, a connection with practical scenarios, and a highlight of its significance.
2. The paper did not include discussion about its relationship with previous work. In particular, how is Eq. (5) fundamentally different from the mean-field parameterization? How does the definition of Eq. (7) relate to the reproducing kernel Hilbert space?
3. Estimating the second layer weights seems to suffer from practicality issue. The estimation is based on the approximation of the expectation based on sampling, where for each sample a minimization problem needs to be solved.
4. Connection between Eq. (29) and the dropout algorithm is not clear.
5. Notations are not explicitly introduced in the main text but deferred to the appendix, making the paper even harder to decode. The definition of $f$ seems to be not clear. In particular, $f$ is defined in Eq. (7), but in Eq. (15) it is not clear why $f$ from Eq. (7) have an explicit dependency on $m$.

**Questions:**

None

---

> ### Author Response · Authors · 2025-11-25
>
> Thank you for the comments.
>
> ### Weaknesses
>
> 1. Integral representations for shallow networks are standard in the literature. Ours differs only slightly as explained in (8), (9), including the use of signed measures.
>
>    With regard to (1), technically, in the continuum limit the index $i$ becomes a continuous variable, say $\iota$ and the sum an integral with respect to some distribution, say $\pi$, so that we obtain $\int a(\iota) \sigma(w(\iota)) \,d\pi(\iota)$. It is not hard to see that without loss of generality we can choose $w = \iota$ and arrive at the formulation in the paper.
>
>    With regard to significance, to the best of our knowledge there are proofs of training algorithms achieving Barron type approximation results. The results of the paper provide a limiting setup where this is possible, which seems to be a major step towards a solution. It remains to show that the results can be matched by finite counterparts. This requires a perturbation analysis and is studied under slightly different conditions in mean-field theories. See also the discussion with Reviewer (CBxT).
>
> 2. Equation (5) is the mean field-parametrization, contained in a section that reviews mean-field theory. A literature overview is given later in Section 1.4. Barron spaces are not Hilbert spaces and pose weaker smoothness conditions than RKHS.
>
> 3. In any practical neural network the outer coefficients are minimized. Since the size of the outer coefficients informs the placement of the inner ones, for theoretical purposes we keep them in an optimized state. Practically, this can be achieved by choosing different learning rates for inner and outer coefficients or by skipping training steps for the inner coefficients.
>
> 4. Can you refine the question? The inner sum in (28) uses only a finite random sub-selection $v_1, \dots, v_m$ of all possible inner coefficients $w_1, \dots, w_{\mathfrak{m}}$, while the remaining ones are dropped. This is the usual definition of dropout.
>
> 5. Are these the intended equation numbers? (15) does neither depend on $m$ nor does it describe $f$. It contains a neural network approximation thereof.
>
>    We agree that the notations would be better placed in the main text. But this would exceed the page limit after adding numerical experiments requested by other reviewers.

---

### Official Review · Reviewer_Behz · 2025-10-31

**Soundness:** 3
**Presentation:** 2
**Contribution:** 3
**Rating:** 4
**Confidence:** 2

**Summary:**

The paper proposes a modified mean-field objective for shallow ReLU networks that directly optimizes the expected sampling error of finite networks rather than the usual continuum loss. Concretely, the authors (i) introduce the new loss with the expectation taken outside the norm; (ii) prove that this loss is convex over admissible weight distributions; (iii) show that any local minimizer enforces an equidistribution condition implying that each sampled neuron contributes with equal magnitude; (iv) obtain Barron-type approximation rates for networks sampled from these minimizers; and (v) connect particle approximations and gradient-descent training to dropout-like regularization. Two variants are developed: an idealized setting with exact outer weights and a practical construction based on averaging optimal outer coefficients conditioned on one neuron.

**Strengths:**

The theory is crisp and tight: the paper proves finite‑width approximation theory in the Barron space, and prove equidistribution property of the minimizer as a necessary optimality condition arising directly from the convex objective, which is an interesting observation. The whole theoretical story is complete.

**Weaknesses:**

While the finite particle approximation algorithm is clearly described, the manuscript includes no experiments to verify the equidistribution, or approximation property in practice.

**Questions:**

Could the authors include a small synthetic experiment that visualizes the learned $\pi$ vs. a classical baseline and measures finite‑width error vs. $m$ under your loss?

---

> ### Author Response · Authors · 2025-11-25
>
> Thank you for the comments. The loss is practically realized by dropout regularization. We've added some preliminary experiments, which confirm equidistribution.

---

### Official Review · Reviewer_Wzry · 2025-10-31

**Soundness:** 3
**Presentation:** 3
**Contribution:** 2
**Rating:** 4
**Confidence:** 3

**Summary:**

This paper studies the approximation theory and optimization landscape of shallow neural networks through a modified mean field perspective. The authors introduce a new loss function that directly optimizes the expected sampling error of finite-width networks, diverging from traditional mean field formulations. The work establishes convexity of the loss with respect to the weight distribution, analyzes the equidistribution of weights at local minima, and demonstrates Barron-type approximation rates under both idealized and practical outer weight choices. The paper further discusses how particle approximations of the new loss naturally result in gradient methods akin to dropout regularization, aiming to bridge mean field insights to practical finite-width training.

**Strengths:**

1. Discussion about best choice of distribution $\pi$: By reformulating the mean field loss so that the expectation is taken outside the norm, the article considers how to choose a distribution to minimize the Barron norm on the right side. This is a subtle but important problem for practical applications.
2. Mathematical rigor: Major theoretical results are clearly stated and appear logically sound. Convexity is proven carefully, and the equidistribution result is both elegant and insightful.
3. Comprehensive literature review: The introduction situates the work well among mean field theory, Barron approximations, landscape/optimization studies, and recent literature.

**Weaknesses:**

1. Lack of convergence rate improvement: Although the paper considers improvements to the distribution \pi, its approximation rate with respect to parameter size is not further improved. However, as the paper points out, under some activation functions [1], the approximation rate can actually be improved.
2. Discussion of completeness issues: The entire paper only focuses on the error in the case of infinite samples. For the case of finite samples, the relationship between approximation error and generalization error is not clear.
3. The algorithm's motivation is unclear: The algorithm proposed in the paper optimizes the distribution of a parameter. Because the corresponding problem is convex, it is guaranteed to be optimal after convergence and to achieve a controllable convergence rate for the objective function. Subsequently, if a specific implementation of the approximation is desired, it is achieved through specific sampling. However, specific sampling does not guarantee specific error control. Therefore, why consider optimizing this loss instead of directly randomly initializing a neural network and performing gradient descent?
4. The computational efficiency of the algorithm: Although the article has discretized the expectation, Eq. (28) shows that it has not discretized the space. Will this dual discretization of expectation and space affect the computational efficiency and its operability?
5. Lack of specific implementation of the algorithm: Although the article proposes a specific algorithm from a theoretical perspective, it does not provide a specific implementation. It would be beneficial if the algorithm's operability and advantages could be demonstrated through experiments.

Reference:
[1] J. W. Siegel and J. Xu. Sharp Bounds on the Approximation Rates, Metric Entropy, and n-Widths
of Shallow Neural Networks. Foundations of Computational Mathematics,2 4(2):481–537, Apr. 2024.

**Questions:**

1. On approximation-rate gains: Your approach improves $\pi$, but the approximation efficiency with respect to parameter size does not improve. Which assumptions or settings in your framework hinder this improvement? Can your analysis and algorithm be tweaked to achieve higher approximations? In particular, the efficiency of parameter approximations is related to the curse of dimensionality, which is why people are interested in barron spaces. Could improving the efficiency of approximations with respect to $\pi$ lead to relevant insights?
2. Regarding the connection with generalization: The entire article only discusses the impact of approximation rate, and both theoretically and experimentally focuses on $L^2$ loss. For the discretized version with finite parameters, the relationship between approximation error and generalization error needs to be discussed.
3. On the motivation for optimizing the “distributional” loss: Since the convexity of the loss guarantees optimality, what concrete advantage does optimizing this loss have over directly training a randomly initialized neural network with gradient descent on the original objective? Is there a principled consistency bridge from the convex distributional optimum to a finite network that preserves risk within a provable tolerance rather than in the sense of expectation?
4. On the discretization and computational efficiency: Eq. (28) discretizes the expectation but keeps the space continuous. In practice, do you also discretize the parameter space (e.g. grid), and if so, how does this affect complexity and error propagation?
5. On practical implementability and evidence: Could you provide a minimal end-to-end implementation showing how the distribution is optimized and then instantiated as a finite model? Even a small-scale experiment (synthetic or standard benchmarks) demonstrating training time, final risk, and sample efficiency vs. a baseline neural network would clarify the operability and benefits of your approach.

---

> ### Author Response · Authors · 2025-11-25
>
> Thank you for your comments.
>
> ### Weaknesses
>
> 1. A higher approximation rate is not expected from improved $\pi$. For comparison, both Monte Carlo sampling from uniform distributions and Markov chain Monte Carlo sampling from adapted distributions have rate $m^{-1/2}$. Likewise, for splines, adaptive placement of inner knots, analogous to $\pi$ in our case, does not improve the (maximal) rate. In both cases, one only achieves the maximal rate conditional under the assumption that the underlying problem is sufficiently nice. Adapting $\pi$ ensures that we can retain the maximal rate with good constants for rougher problems. In the paper this distinction is made by the type of Barron norm, see Appendix C.2 for an example. See also the comments for Reviewer (CBxT)
>
>    With regard to the reference [1], the rate improvements diminish in high dimensions. Therefore, we have decided not to include them, in favour of keeping these first equidistribution results as simple as possible. Incorporating improved rates would be interesting for future work.
>
> 2. The paper allows any Hilbert space norm for the loss. This includes sample losses. Estimation bounds can then be shown by Rademacher complexity results.
>
> 3. Analytically, it is often easier to analyze algorithms in continuous limit cases and consider the finite implementation as a perturbation. This is the common approach in mean field theory. The same strategy is also used in neural tangent kernel arguments, which take the infinite width limit of the derivative $\nabla f (\nabla f)^T$ and consider finite networks as perturbation.
>
> 4. See Item 2. The algorithm itself is gradient descent with dropout type regularization and has similar complexity. We keep the outer weights trained for simplicity. Practically, one can train them with a higher learning rate or skip training steps for the inner weights.
>
> 5. We've added some preliminary numerical experiments before the references.
>
> ### Questions
>
> 1. As a rule of thumb in approximation theory, smoothness allows higher rates and adaption, such as $\pi$ in our case, allows us to keep good rates for more difficult functions $f$. For context, this is the foundation of $hp$ adaptive finite element methods in scientific computing.
>
>    To provide an example, consider a function $f = g(Ux)$ for some wide matrix $U \in \mathbb{c \times d}$. To obtain rate $m^{-1/2}$ with uniform $\pi$ we would need significant dimension dependent Sobolev smoothness $H^{(d+3)/2}$. On the other hand, an adaptive $\pi$ can find the intrinsic low dimensionality and would only require smoothness $H^{(c+3)/2}$.
>
>    Hence, to fully benefit form the dimension independence of Barron results, we have to properly adapt the measure $\pi$. This is exactly what this paper is about, even if the rate $m^{-1/2}$ remains unchanged.
>
>    See [https://arxiv.org/abs/2505.00351](https://arxiv.org/abs/2505.00351) for details.
>
> 2. As other reviewers have noted the paper already has "a large number of appendices containing both background material and proofs". Therefore, it seems fair to leave a thorough analysis of estimation errors for future work.
>
> 3. For training the network weights directly, the loss is not convex, so that convergence to global or good minima is not clear. One can prove convergence under restrictive conditions, e.g. the lazy training regime. But then the inner weights barely move, which is insufficient for Barron type approximation bounds and dimension independence as discussed above.
>
>    The reviewer's question if optimization results for the distribution carry over to finite counterparts as e.g. particle approximations, is very important. It is a major topic in mean-field theories. Adaptions to our setup will be considered in future work.
>
> 4. The algorithm and (28) is an outlook for future work. We expect optimizers for $\pi$ to be stable under perturbations, weather they originate from the sampling indicated in Section 4 or from sampling in space, so that concrete bounds can likely be established.
>
> 5. For the new preliminary experiments, we chose plain dropout regularization. This is slightly simpler than the theoretical approximation $\bar{a}_\pi$, but sufficient to show the theoretical equidistribution properties.

---

### Official Review · Reviewer_CBxT · 2025-11-01

**Soundness:** 3
**Presentation:** 2
**Contribution:** 2
**Rating:** 4
**Confidence:** 3

**Summary:**

The paper studies finite shallow models (neural networks with one hidden layer) and their equidistribution properties from the perspective of Barron norms and discrete mean field theory. First, the paper recalls Barron-type convergence bounds for approximations obtained by sampling the neurons, and discusses the equidistribution property associated with optimal sampling measures (section 3.2). Moreover, the paper examines the quadratic network loss as a function of the sampling measure under assumption of exact outer weights, and shows that the loss is convex and its optima have similar equidistribution and convergence properties (section 3.1). After that the paper proposes a replacement of the theoretical exact outer weights by suitable computable alternatives. It is shown that, up to some perturbation terms, these computable alternatives also provide similar equidistribution and convergence properties (section 3.2). Finally, section 4 provides a preliminary informal discussion connecting the proposed constructions to Wasserstein gradient descent and dropout.

**Strengths:**

**Clarity and quality**

The first half of the paper (sections 1 - 3.1) is very well written. The logical flow is clear, the results are presented as concisely formulated rigorous theorems. However, starting from the averaged outer weights (section 3.2) the exposition was progressively harder to follow for me.

The mathematical level of the paper is on the whole quite high. The authors are very familiar with both classical (e.g., Barron's) and state of the art (mean field, Wasserstein flow etc.) approximation theory of shallow models.

The paper has a large number of appendices containing both background material and proofs of the theorems presented in the main text.


**Originality and significance**

While shallow networks is a more or less well-explored topic and its relevance to state-of-the-art deep models is limited, the particular ideas proposed in the paper - e.g., the approximate outer weights $\\overline{a}_{\\pi}$ - seem to be new and potentially have merit.

**Weaknesses:**

1. The main problem with this work is that it looks unfinished to me. The main new contributions are concentrated in the second half of the paper, but I find the exposition towards the end rather hectic. The results are either not stated as clear theorems, or are stated in terms of some auxiliary quantities (e.g., perturbation terms $\\Delta_k$). The importance of these new theorems (lemma 3.4, theorem 3.5) is not clear to me. The last section 4 is particularly vague and hard to appreciate. None of the presented results looks truly useful or memorable.

2. While the paper aims to address practical finite networks, it presents no actual examples or applications of the developed theory.

**Questions:**

I didn't carefully study the proofs, but there seem to be some minor issues with formulas here and there.
- It seems that lemma C.1 needs some polishing. The first equality in line 914 is probably meant to be an inequality. On the other hand, it's not very clear why the lemma itself is stated in terms of inequalities: if the norm is euclidean and $X_i, X'_i$ are centered i.i.d., then all three expressions in line 903 seem to be connected by equalities using suitable coefficients.  The proof of the lemma and its applicaton to the proof of theorem 2.1 can be slightly clarified.
-  The norm $\\|f\\|_B$ in the seconf part of formula (12) seems to be $|f|_B$; it is identical to the norm in line 231.

---

> ### Author Response · Authors · 2025-11-25
>
> Thank you for the comments.
>
> Although shallow networks are "more or less well-explored", there remain important open problems. To the best of my knowledge, we cannot prove that training algorithms match Barron approximation results. Greedy methods are not yet fully dimension independent. Mean field theories require more than Barron smoothness, usually properties of the initial distribution before training. Unlike Barron smoothness, these assumptions can be significant, e.g. if a uniform initial guess misses unknown intrinsic low dimensionality of the target $f$.
>
> ### Weaknesses
>
> 1. The second part is more technical and therefore also more difficult to understand. For the auxiliary quantities as $\Delta_k$ Appendix B contains some preliminary estimates, but a full analysis is beyond the scope of this paper. The last Section 4 serves as an outlook.
>
>    The equidistribution results shown in the paper are foundational for adaptive approximation methods. They allow us to show approximation rates under weak smoothness assumptions, here of Barron type, less than commonly used in mean field theory. See Appendix C.2 for a very simple example.
>
> 2. We've added some preliminary numerical experiments before the references.
>
> ### Questions
>
> 1. Corrected, modified the proof and replaced inequalities with equalities. We've also updated the constants in all applications. The original variant is from the reference before the lemma and works more generally for type-2 Banach norms.
> 2. Corrected.

---

### Author Response · Authors · 2025-12-03

We would like to address the questions on significance, raised by some reviewers. Optimizers converge to critical points for which the paper shows approximation properties based on equidistribution principles. Throughout approximation theory, the power of a method is measured by two quantities: First approximation rate, here error $\mathcal{O}(m^{-1/2})$ for network width $m$. Second the required regularity of the target function, in our case Barron smoothness. As a rule of thumb, higher smoothness and polynomial degrees lead to higher rates, while adaptivity such as placement of spline knots, or inner weights in this manuscript, allows weaker regularity assumptions. This observation is foundational and the basis of important methods like $hp$ adaptive finite elements. Since the paper addresses the optimization of inner weights, the expected improvements are in the regularity conditions, not the rates. We use Barron smoothness, which is weaker that the requirements in current mean field optimization theory. This seems particularly important for the networks to exploit intrinsic low dimensionality of the target. Hence, the improvements are subtle, but do seem relevant to us.

We've added some preliminary experiments, as requested by the reviewers.

---

### Meta-Review · Area_Chair_T4js · 2026-01-06

**Summary:**

This paper proposes to replace the standard mean-field loss by an expected sampling error, where the authors argue that the new problem has several nice properties, including convexity, equidistribution, and an approximation error bound for local minimizers.

However, the reviewers have concerns about the presentation of the paper overall and the major results in particular. They also questioned if the method proposed can indeed be practical, and requested experiments.

While the reviewers appreciate the promising new theory, they suggested that the paper would benefit from making clear its contributions and relation to prior work/practical applications.

**Reviewer Concerns:**

Some reviewers were concerned about the presentation, e.g., that the work is not finished, and the authors did not clearly explain the importance of their theorems. Some notations are delayed to the appendix, and some motivations of the paper are not clear to the reader.
The reviewers all share concerns about practicality and implementation of the work, and their connection to the theory. While the authors added small experiments with sampling, it is not clear how these practical versions relate to the theory that they proposed. The connection with dropout is not clear.

**Reviewer Scores:**

The reviewers did not respond in the initial stage, they may or may not respond after the authors add experiments, so that specific concern may be resolved, but other concerns remain.

---

### Decision · Program_Chairs · 2026-01-26

Reject